# GAUSS: Graph-Assisted Uncertainty Quantification using Structure and Semantics for Long-Form Generation in LLMs

**Karthik Somayaji NS** [1]   **Yuxuan Yin** [1]   **Peng Li** [1]

## Abstract

In critical domains like clinical reporting, legal analysis, and policy drafting, large language models (LLMs) are increasingly expected to produce extended, fact-rich narratives rather than isolated sentences. Reliable uncertainty quantification in such long-form outputs is crucial. Existing techniques either assign a single confidence score to an entire paragraph or evaluate factual consistency by comparing extracted atomic facts across multiple generations. Some recent approaches represent fact–paragraph relationships using bipartite entailment graphs and derive uncertainty from fact centrality. However, these methods ignore the explicit dependencies among facts within a paragraph and the structural and semantic variation across multiple LLM outputs for the same prompt, missing a key source of uncertainty specific to long-form generation. We propose GAUSS (**G**raph-**A**ssisted **U**ncertainty **Q**uantification using **S**tructure and **S**emantics), a principled framework that models each generated paragraph as a semantic graph of atomic facts and their relations. We posit that uncertainty arises from structural and semantic discrepancies among these graphs across different samples. GAUSS quantifies uncertainty as the expected alignment cost between the semantic graph of an anchor paragraph and those of alternative generations. By capturing both semantic content and structural coherence, GAUSS offers a more interpretable and theoretically grounded measure of uncertainty than coarse, sentence-level scores.

[1]Department of Electrical and Computer Engineering, University of California, Santa Barbara, USA. Correspondence to: Peng Li <lip@ucsb.edu>.

*Proceedings of the 43$^{rd}$ International Conference on Machine Learning*, Seoul, South Korea. PMLR 306, 2026. Copyright 2026 by the author(s).

## 1. Introduction

Large Language Models (LLMs) are increasingly used in high-stakes domains requiring strong factual accuracy [27, 11, 12, 13, 9]. While they excel at fluent long-form generation, their outputs often exhibit hallucinations and inconsistencies. This makes reliable uncertainty quantification essential in critical settings. While recent advances in uncertainty quantification (UQ) have made progress on short-form generation using semantic features, conformal calibration, and entropy-based metrics [10, 14, 24, 4, 25, 8, 17, 19], these methods remain constrained to isolated sentences or atomic facts. Long-form generation, as produced by LLMs is inherently paragraphic in nature: it weaves together multiple atomic facts in a structured, interdependent manner. These facts are not merely co-located; they exhibit logical flow, hierarchical relationships, and latent semantic dependencies. Traditional UQ techniques, which assess uncertainty in isolation or via entropy over discrete outputs [17, 19], struggle to capture such organization. Recent work [34, 15, 35] has sought to extend UQ to long-form generation by decomposing paragraphs into atomic facts, evaluating each via entailment models or consistency checks, and aggregating the results into a paragraph-level score. However, treating facts as independent units ignores the structural coherence that underpins long-form content. Logical flow, contextual dependencies, and the nuanced arrangement of facts are discarded, leading to coarse and potentially misleading uncertainty estimates. Reliable, interpretable UQ for long-form generation demands moving beyond 'bag-of-facts' analyses toward representations that reflect the structure and semantics of entire paragraphs.

To build reliable and interpretable UQ for long-form generation, we argue that uncertainty should be grounded in both the internal organization and semantic meaning of the paragraphs themselves. In this regard, we propose GAUSS (**G**raph-**A**ssisted **U**ncertainty quantification using **S**tructure and **S**emantics for Long Form Generations in LLMs), a principled framework for modeling uncertainty in long-form LLM outputs through graph-based representations (see Figure 1). In GAUSS, each generated paragraph is first decomposed into its constituent atomic facts, which are represented as nodes in a semantic graph. Edges between these

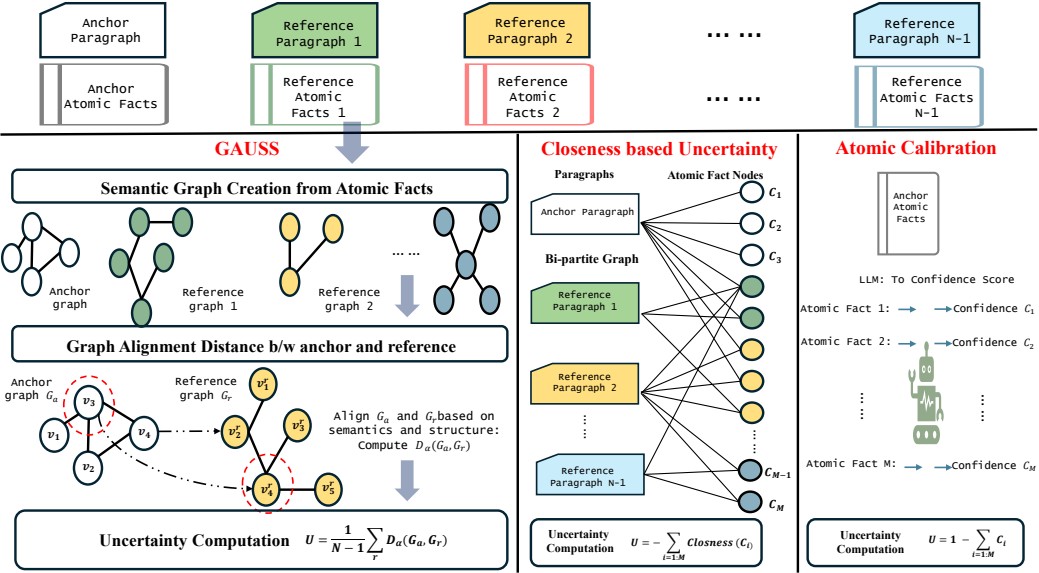

*Figure 1.* GAUSS decomposes paragraphs into atomic facts and represents each as a semantic graph, capturing both factual semantics and inter-fact relations. Uncertainty is computed via graph alignment between an anchor and the remaining reference graphs. In contrast, [16] uses a single bipartite graph and centrality over fact–paragraph entailments, while [34] relies on external LLMs and ignores intra-paragraph structure.

nodes encode pairwise semantic relationships, capturing dependencies among factual elements. This graph-based abstraction serves two key roles: (i) it preserves the symbolic structure inherent in long-form text, and (ii) it embeds the semantic content of individual atomic facts. To quantify uncertainty, GAUSS compares the semantic graph of a generated paragraph against those of other candidate generations, measuring structural and semantic deviations using a graph alignment distance. In doing so, GAUSS offers a structure-aware approach to UQ in long form generation. A recent graph-based approach [16] represents atomic fact–paragraph interactions using a single bipartite entailment graph and quantifies fact-level uncertainty via graph centrality metrics such as closeness (see Figure 1). [16] model fact level uncertainty through relationships between facts and paragraphs across generations, without explicitly capturing the rich internal organization/ coherence of atomic facts in individual paragraphs. In contrast, GAUSS takes a fundamentally different approach: it constructs a separate semantic graph for each paragraph, with nodes representing atomic facts and edges encoding semantic and structural dependencies. This per-paragraph semantic graph modeling enables GAUSS to assess uncertainty at generated paragraph level by directly comparing structure and meaning across generations. Furthermore, unlike [16], GAUSS offers theoretical guarantees, and extends naturally to atomic-level uncertainty (GAUSS-atomic), content filtering applications etc (Appendix Sections A, B, C). [26, 5] leverage graph structures for reasoning-based uncertainty, using deducibility or explanation graphs with conformal prediction or graph-edit distances. GAUSS instead targets long-form

outputs by aligning semantic graphs of atomic facts. Thus, our key contributions are:

1. We introduce a semantic-graph representation that simultaneously encodes the meaning of each atomic fact and the relational dependencies among them to capture both content and structure in long-form paragraphs.

2. We propose GAUSS, a structure and semantics-aware framework for uncertainty quantification in long-form generation, which estimates uncertainty via fused Gromov–Wasserstein graph alignment distance between semantic graphs.

3. We theoretically establish the Lipschitz continuity of both the graph alignment distance and the resulting uncertainty measure under semantic and structural perturbations, ensuring robustness to small graph structure and semantic embedding variations in generated text.

4. We derive exponential convergence bounds for the uncertainty measure in terms of the number of sampled paragraphs and the consistency of the generating LLM, demonstrating that reliable uncertainty estimates can be obtained with modest sample sizes.

## 2. Motivation

We posit that uncertainty in long-form generation should reflect the variability in how atomic factual units (atomic facts) within the generated paragraphs are represented and semantically interconnected. To operationalize this concept, consider a scenario wherein an LLM generates multiple

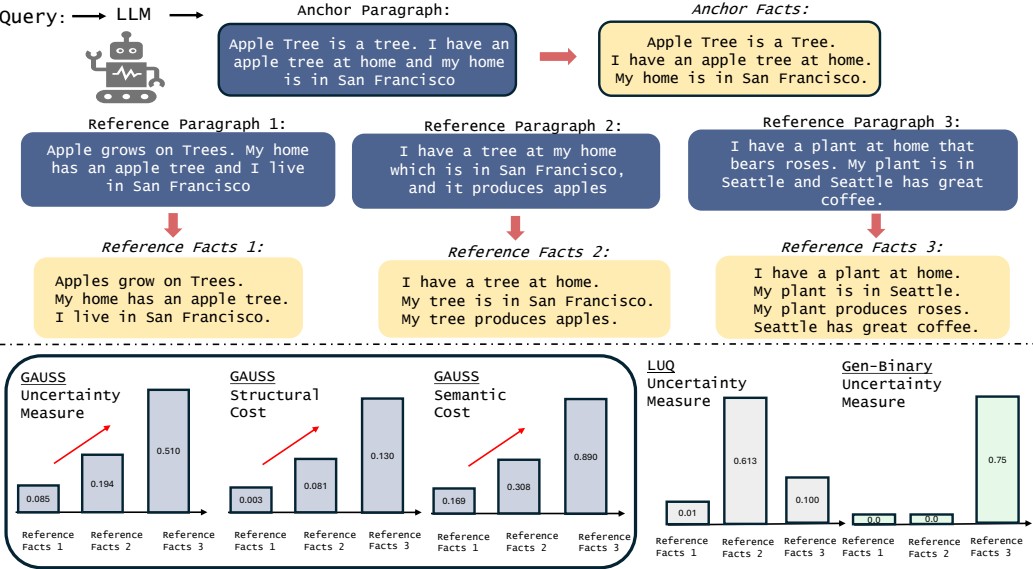

*Figure 2.* An anchor paragraph and three references diverge in semantics and structure. `GAUSS` uses graph-based structural + semantic costs to produce steadily increasing uncertainty (left), whereas semantic-only baselines (right) give inconsistent scores, highlighting the need for joint structure–semantics comparison.

paragraphs in response to the same query for a long-form answer. We can designate one paragraph as the anchor and all others as reference paragraphs and decompose each generated paragraph into its constituent atomic facts. Consequently, uncertainty in paragraph generation can be assessed by examining the structural and semantic discrepancies between the atomic facts of the anchor and reference paragraphs.

To analyze such differences rigorously, we represent each paragraph as a semantic graph, where nodes encode the meaning of individual atomic facts, while edges quantify semantic relationships between them. This structured representation enables us to capture not only the content of a paragraph but also the relational fabric that holds it together. By comparing these graphs across generations, we obtain an interpretable measure of uncertainty, one that reflects both the semantic variation and organizational shifts inherent to long-form language generation.

Figure 2 shows three reference paragraphs that have been constructed so as to gradually diverge from an anchor in both meaning and structure. Methods like LUQ [35] and Gen-Binary [34] treat long-form uncertainty purely as semantic agreement : either by averaging entailment scores over atomic facts or by querying another LLM for fact support (between anchor facts and other reference paragraphs), thus overlooking how the paragraphs' symbolic structure interacts with their content. In contrast, GAUSS encodes each paragraph as a graph, jointly capturing the inherent structure (atomic fact interdependices) and semantics (atomic fact semantics). By examining the three scenarios in Figure 2, we highlight how this graph-based comparison more faithfully

quantifies uncertainty.

- **Reference Facts 1:** The reference facts closely mirror the anchor facts with minor wording differences. Here, our approach correctly identifies slight structural and semantic variations, consistently demonstrated across comparative methods (GAUSS , Gen-Binary[34], LUQ[35]).

- **Reference Facts 2:** These facts retain overall semantic congruence with the anchor, but differ significantly in their organization, lacking direct one-to-one correspondence with the anchor facts. Unlike approaches relying solely on aggregate semantic similarity, GAUSS explicitly captures variations in generation style and structure, yielding a higher uncertainty score. Conversely, Gen-Binary underestimates uncertainty and fails to recognize different organization style as it solely relies on just the overall semantics.

- **Reference Facts 3:** This scenario introduces significant alterations in both semantics and structure (through contradictory information and extraneous content) in comparison to the anchor facts. Our method accurately identifies these substantial deviations, reflecting increased structural and semantic costs and thereby a higher uncertainty measure.

This example demonstrates our method's sensitivity to nuanced variations in paragraph semantics, structure, and style. In contrast, Gen-Binary [34], which focuses solely on overall semantics, fails to detect organizational differences, overlooking uncertainties evident in reference facts 2. Similarly,

LUQ [35], by disregarding structural organization, does not accurately capture the progression of uncertainty across reference facts. Unlike these methods, GAUSS consistently reflects a monotonic increase in uncertainty across the progressively varying reference paragraphs, emphasizing the need for a structure and semantic aware uncertainty quantification technique for long-form LLM generation.

## 3. Background

Although it has not yet been applied to uncertainty quantification in long-form generations by LLMs, graph-based optimal transport (OT) [20, 30] offers a principled framework for comparing structured data such as graphs, by aligning the nodes of two graphs based on both structural relationships and feature-level similarity. While classical OT aligns distributions over Euclidean spaces, graph alignment requires accounting for relational dependencies between nodes.

Let $G_1 = (V_1, E_1, C_1, \ell_f)$ and $G_2 = (V_2, E_2, C_2, \ell_f)$ be two graphs, where $V_i$ and $E_i$ represent the nodes and edges of the graphs respectively. $C_1 \in \mathbb{R}^{|V_1| \times |V_1|}$ and $C_2 \in \mathbb{R}^{|V_2| \times |V_2|}$ are structure matrices encoding pairwise node relations (e.g., shortest path, adjacency etc), and $\ell_f$ is the node feature mapping function. The goal in graph based OT is to align the two graphs $G_1$ and $G_2$ based on both the structural and semantic similarity. The *fused Gromov–Wasserstein distance* (graph alignment distance) $D_\alpha$ [30, 33], compares both structure and features via:

$$
\begin{aligned}
D_\alpha(G_1, G_2) = \min_{\pi \in \Pi} \sum_{i,j,k,\ell} &\Big[ (1-\alpha)\, m(\ell_f(i), \ell_f(j)) \\
&+ \alpha \left| C_1(i,k) - C_2(j,\ell) \right| \Big] \pi_{ij} \pi_{k\ell},
\end{aligned}
$$
$$
\text{s.t } \Pi \triangleq \left\{ \pi \in [0,1]^{|V_1| \times |V_2|} \;\Big|\; \sum_{j=1}^{|V_2|} \pi_{ij} = 1,\; \sum_{i=1}^{|V_1|} \pi_{ij} = 1 \right\}.
$$
$$(3.1)$$

where $m(\cdot, \cdot)$ is a feature cost function (e.g., cosine distance), and $\alpha \in [0,1]$ trades off structural and feature alignment. The coupling $\pi$ can be seen as the mapping from the nodes of $G_1$ to nodes of $G_2$.

## 4. Graph Based Uncertainty Quantification for Long Form Generation

### 4.1. Overview of the Long Form Generation Uncertainty Estimation Framework

We begin by outlining the core pipeline of GAUSS before detailing each component of the framework.

To quantify the long-form uncertainty associated with a given query $q$ and language model $\mathcal{M}$, we first sample $N$ independent paragraph-length responses:

$$
\{P_i\}_{i=1}^N \sim \mathcal{M}(q).
$$

From these, we designate one paragraph $P_a$ as the *anchor*, and treat the remaining $N-1$ paragraphs $\{P_r\}_{r \neq a}$ as *references*. Each paragraph $P_i$ is decomposed into a set of atomic facts $\mathcal{F}_{P_i}$ using a factual decomposition model $\mathcal{M}_{\mathrm{atomic}}$, and is subsequently represented as a semantic graph $G_i = (V_i, E_i, C_i, \ell_f)$, where nodes encode the atomic facts and edges capture semantic dependencies.

To assess the variability of generations around the anchor, we compute the alignment distance $D_\alpha(G_a, G_r)$ between the anchor graph $G_a$ and each reference graph $G_r$. The final uncertainty score for query $q$ is given by the mean alignment cost:

$$
U(q) = \frac{1}{N-1} \sum_{r \neq a} D_\alpha(G_a, G_r). \tag{4.1}
$$

Intuitively, $U(q)$ captures the average structural and semantic deviation between the anchor and reference paragraphs—higher values indicate greater generation variability and, thus, higher uncertainty. We describe the construction of semantic graphs in Section 4.2, the computation of alignment distances in Section 4.3, and the theoretical properties of the uncertainty measure in Section 4.4. A full overview of the GAUSS pipeline is provided in Algorithm 1.

### 4.2. Representing Paragraphs as Semantic Graphs

To capture both the semantic content and the internal relational structure within a paragraph $P_i$, we represent it as a semantic graph $G_i = (V_i, E_i, C_i, \ell_f)$. The construction of the semantic graph proceeds in three stages:

**Step 1: Atomic Fact Extraction.** We begin by decomposing paragraph $P_i$ into its constituent atomic facts:

$$
\mathcal{F}_{P_i} = \mathcal{M}_{\mathrm{atomic}}(P_i) = \{f_1, f_2, \ldots, f_{n_i}\},
$$

using a prompted model $\mathcal{M}_{\mathrm{atomic}}$ following prior work [32, 22, 34, 35]. Each $f_k$ denotes a standalone factual statement derived from $P_i$. Further implementation details for prompting $\mathcal{M}_{\mathrm{atomic}}$ are provided in the Appendix L.

**Step 2: Node Construction and Semantic Embedding.** Each atomic fact $f_k \in \mathcal{F}_{P_i}$ is treated as a node $v_k$ in the vertex set $V_i$, so that $V_i = \{v_1, \ldots, v_{n_i}\}$. To encode the semantic meaning of each node, we use a sentence embedding model $\mathcal{M}_{\mathrm{sentence}}$ to compute:

$$
\ell_f(v_k) = \mathcal{M}_{\mathrm{sentence}}(f_k) \in \mathbb{R}^d.
$$

**Step 3: Structural Matrix Construction.** To encode inter-fact dependencies, we define the structure matrix $C_i \in \mathbb{R}^{n_i \times n_i}$ using pairwise semantic distance:

$$C_i(k, \ell) = 1 - \cos\big(\ell_f(v_k), \ell_f(v_\ell)\big),$$

where $\cos(\cdot, \cdot)$ denotes cosine similarity between embeddings. Higher values in $C_i(k, \ell)$ correspond to a weaker semantic affinity between facts $f_k$ and $f_\ell$. While we adopt semantic distance here, other graph-based metrics such as graph kernels may also be more generally used. We explore such variants in Appendix D.

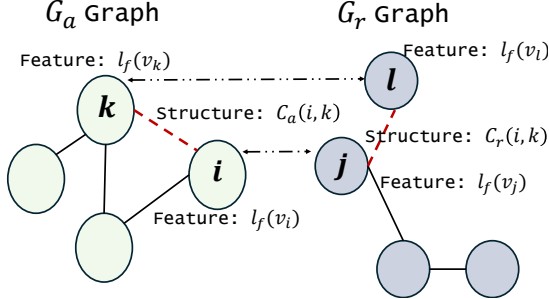

*Figure 3.* The structural and feature cost while aligning graph $G_a$ and reference graph $G_r$.

The resulting graph $G_i = (V_i, E_i, C_i, \ell_f)$ serves as a compact, interpretable representation of $P_i$ and captures both the meaning of each atomic fact and their mutual relationships, setting the stage for principled alignment across paragraph representations.

## 4.3. Computing Alignment Distance Between Semantic Graphs

In this subsection, we detail the computation of the alignment distance $D_\alpha(G_a, G_r)$ between the anchor graph $G_a$ and a reference graph $G_r$, with $n_a$ and $n_r$ nodes respectively. The process involves constructing the semantic cost matrix $M^r$ and the structural cost tensor $L^r$, which jointly define the alignment objective. An illustration is provided in Figure 3.

**Semantic Cost Matrix $M^r$.** The semantic cost matrix $M^r \in \mathbb{R}^{n_a \times n_r}$ encodes the dissimilarity at the feature level between the atomic facts in the anchor graph $G_a$ and those in the reference graph $G_r$. For each pair of nodes $v_i \in G_a$ and $v_j \in G_r$, the cost entry $M^r[i, j]$ is defined as:

$$M^r[i, j] = 1 - \cos\big(\ell_f(v_i), \ell_f(v_j)\big) \cdot \mathcal{M}_{\text{entail}}\big(\ell_f(v_i), P_r\big) \tag{4.2}$$

where $\mathcal{M}_{\text{entail}}(\ell_f(v_i), P_r) \in \{0, 1\}$ is the output of model that determines whether the atomic fact $\ell_f(v_i)$ is supported by the reference paragraph $P_r$. Implementation details and prompting strategies for $\mathcal{M}_{\text{entail}}$ are provided in Appendix L.

This formulation combines both local and global signals: the cosine similarity reflects the local semantic alignment between atomic facts, while the entailment signal serves as a global gating mechanism—suppressing alignment when the reference paragraph lacks the corresponding factual support. Together, they ensure that only semantically and contextually consistent node alignments are favored in the downstream graph matching process.

**Structural Cost Tensor $L^r$.** The structural cost tensor $L^r \in \mathbb{R}^{n_a \times n_a \times n_r \times n_r}$ measures alignment consistency between the internal topologies of the two semantic graphs. Recall that each structure matrix $C_a$ or $C_r$ is computed as:

$$C(i, k) = 1 - \cos\big(\ell_f(v_i), \ell_f(v_k)\big),$$

which reflects the semantic dissimilarity between atomic fact embeddings within a graph. Given this, we define the structural cost tensor as:

$$L^r[i, j, k, \ell] = \big| C_a(i, k) - C_r(j, \ell) \big|, \tag{4.3}$$

which penalizes mismatches in the relative connectivity of node pairs $(i, k)$ in $G_a$ and $(j, \ell)$ in $G_r$.

We also explore other flexible notions of paragraph structure (like **causal dependency**) in Appendix J.

**Final Alignment Distance.** With the semantic cost matrix $M^r$ and the structural cost tensor $L^r$ defined, the alignment distance between $G_a$ and $G_r$, $D_\alpha(G_a, G_r)$ is computed as:

$$D_\alpha(G_a, G_r) = \min_{\pi \in \Pi} \sum_{i,j,k,\ell} \Big[ (1 - \alpha) M^r[i, j] + \alpha L^r[i, k, j, \ell] \Big] \pi_{ij} \pi_{k\ell}.$$

where $\pi$ is a stochastic map which aligns nodes in $G_a$ to nodes in $G_r$. This alignment distance reflects the closest possible semantic similarity and global structural coherence between the anchor and reference graphs through the mapping $\pi$.

## 4.4. Theoretical Properties of the Proposed Uncertainty Measure

In this section, we establish theoretical guarantees for the proposed uncertainty measure $U(q)$, defined in (4.1) as the mean alignment cost between an anchor graph $G_a$ and reference graphs $\{G_r\}_{r \neq a}$ via the distance $D_\alpha$. We first prove the Lipschitz continuity of $D_\alpha$ in Lemma 4.3, and extend it to $U(q)$ in Theorem 4.4, ensuring robustness to semantic and structural perturbations. Theorem 4.5 further establishes exponential convergence of $U(q)$ to its expectation, governed by the number of reference samples and the LLM's consistency, thus offering practical guidance on the sampling budget required for stable uncertainty estimation.

**Algorithm 1** GAUSS

---

**Require:** Query $q$, language model $\mathcal{M}$, number of samples $N$,
  trade-off parameter $\alpha$
1: **Sample Paragraphs:** Generate $N$ long-form responses
  $\{P_i\}_{i=1}^N \sim \mathcal{M}(q)$
2: **Select Anchor:** Choose one sample $P_a$ as the anchor para-
  graph; set remaining $\{P_r\}_{r \neq a}$ as reference paragraphs
3: **for** each paragraph $P_i$ **do**
4:   Extract atomic facts $\mathcal{F}_{P_i}$
5:   Construct semantic graph $G_i = (V_i, E_i, C_i, \ell_f)$
6: **end for**
7: **for** each reference graph $G_r$, where $r \neq a$ **do**
8:   Compute alignment distance $\mathrm{D}_\alpha(G_a, G_r)$
9: **end for**
10: **Expectation:** Compute average alignment cost $U(q) = \frac{1}{N-1} \sum_{r \neq a} \mathrm{D}_\alpha(G_a, G_r) \, U(q)$

---

**Remark 4.1.** The following results provide continuity and convergence guarantees for the uncertainty measure $U(q)$, grounded in graph representations of generated paragraphs. More generally, these properties extend to uncertainty quantification in graph generation, which we formalize below.

Suppose the anchor graph $G_a$ has $n_a$ nodes and a reference graph $G_r$ has $n_r$ nodes. We define the *feature cost matrix* $M^r \in \mathbb{R}^{n_a \times n_r}$ by $M^r[t, h] = m\big(\ell_f(t), \ell_f(h)\big)$, which measures the semantic dissimilarity between node $t \in G_a$ and node $h \in G_r$. We also define the *structural cost tensor* $L^r \in \mathbb{R}^{n_a \times n_a \times n_r \times n_r}$ as $L^r[i, k, j, \ell] = |C_a(i, k) - C_r(j, \ell)|$, where $C_a$ and $C_r$ are the structure matrices of $G_a$ and $G_r$, respectively. This tensor captures discrepancies in pairwise structural relations across the two graphs. For our application to long-form generation, the specific definitions of $M^r$ and $L^r$ are provided in Section 4.3. Together, they fully specify the inputs to the alignment distance $\mathrm{D}_\alpha(M^r, L^r)$ used in computing uncertainty.

**Remark 4.2.** For any fixed pair of graphs $G_a$ and $G_r$, we use the notation $\mathrm{D}_\alpha(G_a, G_r)$ interchangeably with $\mathrm{D}_\alpha(M^r, L^r)$, where $M^r$ and $L^r$ are the corresponding feature cost matrix and structural cost tensor derived from $G_a$ and $G_r$. Thus $\mathrm{D}_\alpha$ - *graph alignment distance*, can be alternatively written as:

$$\mathrm{D}_\alpha(M^r, L^r) = \min_{\pi \in \Pi} \sum_{i,j,k,\ell} \Big[ (1 - \alpha) \, M^r[i, j]$$
$$+ \alpha \, L^r[i, k, j, \ell] \Big] \pi_{ij} \pi_{k\ell}.$$

**Lemma 4.3** (Lipschitz Continuity of Alignment Distance). *The alignment distance $\mathrm{D}_\alpha(M^r, L^r)$ is Lipschitz continuous with respect to both the feature cost matrix $M^r$ and the structural cost tensor $L^r$. That is for any $(M^r, L^r), (\widetilde{M^r}, \widetilde{L^r})$,*

$$\left| \mathrm{D}_\alpha(M^r, L^r) - \mathrm{D}_\alpha(\widetilde{M^r}, \widetilde{L^r}) \right| \leq (1 - \alpha) \| M^r - \widetilde{M^r} \|_\infty$$
$$+ \alpha \| L^r - \widetilde{L^r} \|_\infty.$$

**Theorem 4.4** (Lipschitz Continuity of the Proposed Uncertainty Measure). *Let $U(q)$ denote the uncertainty score for query $q$ with respect to a generated graph $G_a$ as the anchor, and $N - 1$ independently generated graphs $\{G_r\}_{r \neq a}$ as reference graphs, defined as*

$$U(q) = \frac{1}{N - 1} \sum_{r \neq a} \mathrm{D}_\alpha(M^r, L^r), \qquad (4.4)$$

*where $(M^r, L^r)$ are the semantic cost matrices and structural cost tensors between $G_a$ and $G_r$. Then $U(q)$ is Lipschitz continuous with respect to the collection $\{M^r, L^r\}_{r \neq a}$.*

**Theorem 4.5** (Convergence of the Uncertainty Measure). *Under general boundedness assumptions on the alignment distances, the uncertainty score $U(q)$ defined in Eqn (A.2) exponentially converges around the true mean $\mathbb{E}[U(q)]$, specifically:*

$$\mathbb{P}\left[ |U(q) - \mathbb{E}[U(q)]| > \epsilon \right] \leq 2 \exp\left( -\frac{2(N - 1)\epsilon^2}{D^2} \right),$$

*for any $\epsilon$ and some constant $D > 0$ that depends on the graph generation inconsistency.*

**Implication of Theorems 4.4 & 4.5:** The Lipschitz property of $U(q)$ ensures that small semantic or structural perturbations in input graphs lead to proportionally bounded changes in the uncertainty score, making the uncertainty measure robust to noise in the structure extraction or feature embedding process. We provide a practical illustration of the robustness of the uncertainty measure to the embedding process in Appendix E. Theorem 4.5 shows that $U(q)$ concentrates exponentially around its expectation, enabling reliable uncertainty estimation from a modest number of reference samples $N - 1$. Furthermore, for LLMs with low graph generation inconsistency factor $D$, $U(q)$ converges even faster, thus, requiring fewer samples. We provide illustrations of the convergence behavior and inconsistency factor $D$, with two representative LLMs in Appendix G. We provide proofs of the above theorems in Appendix A. We also provide a discussion on the computational cost and runtime of GAUSS in Appendix I.

## 5. Experiments

We evaluate our framework by comparing graph-based uncertainty to factual correctness across three benchmarks and multiple LLMs. [*Datasets*] Specifically, we assess performance on: (1) 183 biography prompts from Bios [22], verified via Wikipedia; (2) 500 open-ended queries from LongFact [32]; and (3) 500 entity-centric samples from the WildHallu chatbot corpus [36]. [*Calibration*] We follow standard practice [34, 35, 15] in evaluating uncertainty by correlating it with the truthfulness of the generated paragraph. We posit that an LLM producing structurally and

*Table 1.* Uncertainty metrics (SC↓, PC↓, UCCE↓, QCCE↓) for different methods across three datasets.

| Method / Model | Bios | | | | LongFact | | | | WildHallu | | | |
|---|---|---|---|---|---|---|---|---|---|---|---|---|
| | SC↓ | PC↓ | UCCE↓ | QCCE↓ | SC↓ | PC↓ | UCCE↓ | QCCE↓ | SC↓ | PC↓ | UCCE↓ | QCCE↓ |
| **falcon-7b-instruct** | | | | | | | | | | | | |
| LUQ | 0.0943 | 0.0555 | 0.2261 | 0.2372 | -0.4067 | -0.4135 | 0.2865 | 0.2600 | -0.3385 | -0.2913 | 0.1295 | **0.1864** |
| Gen-Binary | -0.1535 | -0.1313 | 0.2110 | 0.1896 | -0.6443 | -0.6828 | 0.2020 | 0.2926 | **-0.7628** | **-0.7730** | 0.1654 | 0.2167 |
| Dis-Rating | 0.0134 | 0.0098 | 0.1645 | 0.2092 | -0.0479 | -0.0693 | **0.1786** | 0.3201 | 0.0418 | 0.0492 | **0.1036** | 0.3223 |
| Dis-Single | -0.1571 | -0.2625 | **0.1388** | 0.3349 | 0.0164 | 0.0991 | 0.2216 | 0.4512 | 0.0102 | 0.0240 | 0.1530 | 0.5153 |
| Centrality | -0.2670 | -0.0683 | 0.1316 | 0.1769 | -0.0795 | -0.1813 | 0.2162 | 0.3044 | -0.5441 | -0.4955 | 0.1683 | 0.2385 |
| GAUSS | **-0.4118** | **-0.3321** | 0.168 | **0.1845** | **-0.6555** | **-0.6915** | 0.1817 | **0.2199** | -0.7565 | -0.7616 | 0.1470 | 0.1978 |
| **llama3-8b-instruct** | | | | | | | | | | | | |
| LUQ | -0.0395 | -0.0546 | 0.1053 | 0.1316 | -0.0894 | -0.0452 | 0.2397 | 0.2959 | -0.4682 | -0.3939 | 0.2160 | 0.2723 |
| Gen-Binary | -0.5986 | -0.5909 | 0.1582 | 0.1256 | -0.3731 | -0.3974 | 0.1919 | 0.2726 | -0.6567 | -0.6995 | 0.2198 | 0.2740 |
| Dis-Rating | -0.6495 | -0.5372 | **0.0812** | **0.1058** | -0.2931 | -0.3204 | 0.2052 | 0.2610 | -0.6558 | -0.6779 | 0.2314 | 0.2747 |
| Dis-Single | -0.5143 | -0.5060 | 0.1156 | 0.1385 | -0.4184 | -0.3494 | 0.1889 | 0.2938 | -0.6597 | -0.6936 | 0.2704 | 0.2911 |
| Centrality | -0.6423 | -0.5429 | 0.1086 | 0.1608 | -0.3430 | -0.3379 | 0.2119 | 0.2573 | -0.5833 | -0.5971 | 0.2314 | 0.1883 |
| GAUSS | **-0.7066** | **-0.708** | 0.1035 | 0.1426 | **-0.4433** | **-0.4505** | 0.1613 | 0.2535 | **-0.6808** | **-0.7144** | 0.2048 | 0.2624 |
| **qwen2-7b-instruct** | | | | | | | | | | | | |
| LUQ | -0.0658 | -0.0698 | 0.1346 | **0.0857** | -0.2138 | -0.2378 | 0.2580 | 0.2729 | -0.3944 | -0.3830 | 0.2324 | 0.2838 |
| Gen-Binary | -0.5950 | -0.5954 | 0.1487 | 0.1232 | -0.4542 | -0.4781 | **0.1206** | 0.2732 | -0.6414 | -0.6045 | 0.2097 | 0.3080 |
| Dis-Rating | -0.4611 | -0.4535 | **0.1079** | 0.1336 | -0.3501 | -0.4355 | 0.1785 | 0.2656 | -0.5805 | -0.6726 | 0.1867 | 0.2220 |
| Dis-Single | -0.5063 | -0.5204 | 0.1144 | 0.1027 | -0.4463 | -0.4721 | 0.1695 | 0.2806 | -0.6554 | -0.6367 | 0.1826 | 0.2534 |
| Centrality | -0.6342 | -0.5023 | 0.0545 | 0.1490 | -0.4005 | -0.3164 | 0.2306 | 0.3020 | -0.5069 | -0.4956 | 0.2212 | 0.3081 |
| GAUSS | **-0.6915** | **-0.7114** | 0.1606 | 0.1505 | **-0.4979** | **-0.5233** | 0.1403 | **0.2362** | **-0.7072** | **-0.7154** | **0.1798** | **0.2212** |
| **qwen2-57b-instruct** | | | | | | | | | | | | |
| LUQ | 0.0491 | -0.0308 | 0.1263 | 0.0850 | -0.1120 | -0.0871 | 0.3133 | 0.3507 | -0.3641 | -0.3495 | 0.2900 | 0.3238 |
| Gen-Binary | -0.6093 | -0.5975 | 0.1346 | 0.0907 | -0.3328 | -0.3226 | 0.2476 | 0.3715 | -0.6434 | -0.6681 | 0.2400 | 0.2953 |
| Dis-Rating | -0.6547 | -0.5232 | **0.0676** | **0.0723** | -0.3255 | -0.3776 | 0.2038 | 0.3394 | -0.6084 | -0.6693 | 0.2603 | 0.2503 |
| Dis-Single | -0.6470 | -0.6251 | 0.1230 | 0.1732 | -0.3505 | -0.3428 | 0.1934 | 0.3340 | -0.5906 | -0.6211 | 0.2798 | 0.2154 |
| Centrality | -0.5782 | -0.5310 | 0.0844 | 0.1889 | -0.2545 | -0.1478 | 0.2015 | 0.3252 | -0.3728 | -0.4409 | 0.2448 | 0.3499 |
| GAUSS | **-0.6991** | **-0.7018** | 0.1328 | 0.1092 | **-0.4226** | **-0.4615** | **0.1844** | 0.3111 | **-0.6852** | **-0.7032** | **0.2061** | **0.2043** |
| **mistral-7b-instruct** | | | | | | | | | | | | |
| LUQ | 0.0541 | 0.0485 | 0.1302 | 0.1914 | -0.1762 | -0.1768 | 0.2548 | 0.2876 | -0.2108 | -0.1892 | 0.2946 | 0.3382 |
| Gen-Binary | -0.5803 | -0.6002 | 0.1613 | 0.1756 | -0.4138 | -0.4538 | 0.2631 | 0.2951 | -0.6889 | -0.7552 | 0.1774 | **0.2238** |
| Dis-Rating | -0.4683 | -0.4083 | 0.1539 | 0.1526 | -0.3548 | -0.3960 | 0.2511 | 0.3349 | -0.5628 | -0.6362 | **0.1200** | 0.2544 |
| Dis-Single | -0.1704 | -0.1286 | **0.0981** | 0.1769 | 0.0818 | 0.0281 | **0.1912** | 0.2880 | -0.0899 | -0.1458 | 0.1720 | 0.2970 |
| Centrality | -0.5181 | -0.5456 | 0.0421 | 0.1374 | -0.4118 | -0.3984 | 0.2679 | 0.2645 | -0.5702 | -0.5549 | 0.2750 | 0.3154 |
| GAUSS | **-0.6643** | **-0.6766** | 0.1443 | **0.1407** | **-0.4408** | **-0.4678** | 0.2525 | **0.2672** | **-0.6949** | **-0.7584** | 0.1784 | 0.2310 |

semantically divergent responses for the same query is more likely to generate low-veracity content. Thus, a high uncertainty score from GAUSS should correspond to lower factuality. For each query, we compute the factuality of the anchor paragraph $P_a$ at the atomic fact level. To accomplish this, we follow the SAFE framework [32]: each atomic fact is paired with the top-ranked web search snippets given the query. Each atomic fact is then presented to the verifier model (Qwen2-32B-Instruct [3]) alongside the web search snippet. The verifier classifies whether the fact is supported, and we average these binary outcomes to yield a paragraph-level factuality score. We employ $N = 20$ paragraph generations in Algorithm 1.

[**Baseline Methods**] We compare GAUSS against five representative baselines for long-form uncertainty estimation. Sampling-based methods generate multiple paragraphs and assess uncertainty via inter-sample disagreement. LUQ [35]

(atomic variant) computes entailment probabilities between each atomic fact in the anchor and all reference paragraphs using an MNLI model; uncertainty is defined as one minus the average confidence. Gen-Binary [34] uses an LLM to assess factual support for each atomic fact across references, averaging these consistency scores to yield paragraph-level uncertainty. Single-sample methods operate on a single paragraph and query the generating LLM for internal confidence. Dis-Single [34] prompts the LLM for binary truth labels per fact, while Dis-Rating elicits 0–10 confidence scores. In both, uncertainty is computed as one minus the average per-fact confidence. However, all these baselines treat atomic facts independently, ignoring the structural and semantic relationships that GAUSS explicitly models. We also compare GAUSS with Centrality [16]. To compute the uncertainty metric from [16], we use the negative of the closeness centrality score, following the improved formulation by Wasserman and Faust. A bipartite graph is constructed

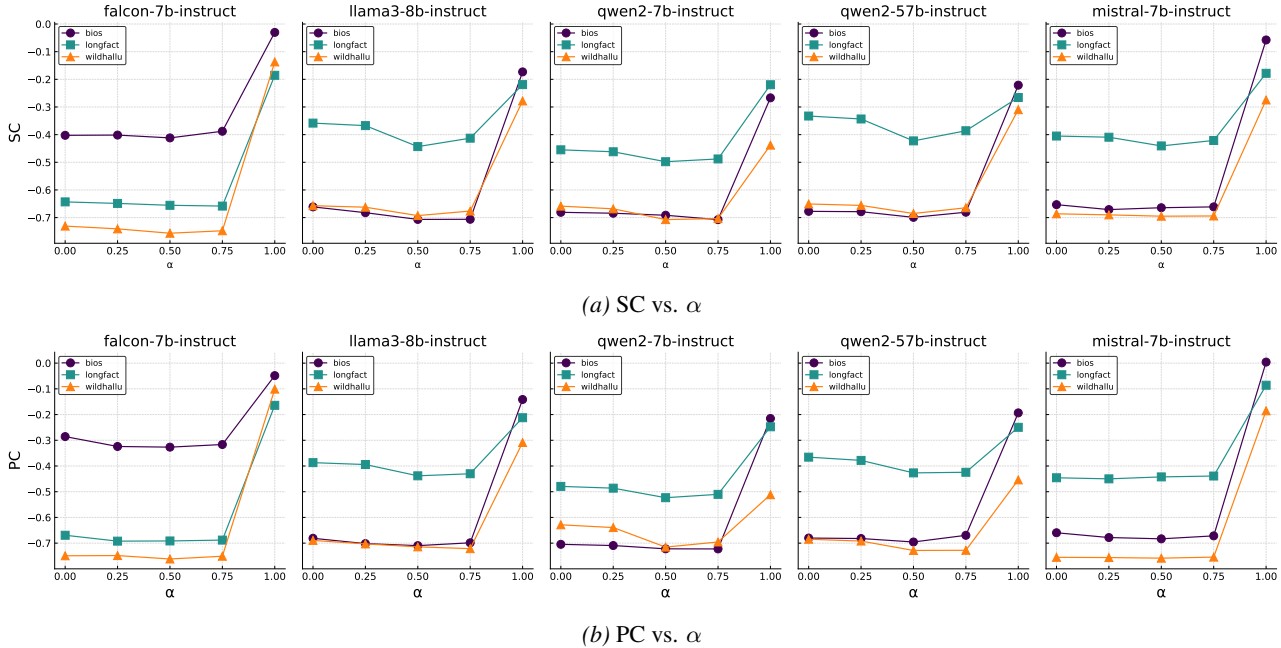

*(a)* SC vs. $\alpha$

*(b)* PC vs. $\alpha$

*Figure 4.* Ablation of varying $\alpha$ from 0 (only semantic cost) to 1 (only structural cost). The plots show the effect of $\alpha$ on Spearman correlation (SC) and Pearson correlation (PC). Combining structural and semantic costs is necessary for lower SC and PC values.

over 10 generated responses and their constituent claims. To obtain a paragraph-level score, we calculate closeness for all claims within a designated anchor paragraph and take the negative average as the final uncertainty estimate. [**Evaluation Metrics**] To measure the effectiveness of the uncertainty estimates, we use multiple evaluation metrics. Spearman correlation (SC) and Pearson correlation (PC) are used to measure monotonic and linear correlations, respectively, between uncertainty scores and factuality labels produced by [22, 32]. We also report Uniform Continuous Calibration Error (UCCE) [34] which measures the average deviation between predicted uncertainty and $1-$ ground-truth factuality across equally spaced bins: $\text{UCCE} = \sum_{m=1}^{M} \frac{|B_m|}{N} \left| \frac{1}{|B_m|} \sum_{i \in B_m} \hat{y}_i - \frac{1}{|B_m|} \sum_{i \in B_m} y_i \right|$, where $B_m$ is the $m$-th bin, $\hat{y}_i$ the normalized predicted uncertainty in the bin, and $y_i$ is the normalized $1-$ factuality score in the bin. We additionally report Quantile Continuous Calibration Error (QCCE), a variant of UCCE that uses quantile-based bins to ensure equal sample sizes. Lower UCCE and QCCE indicate better calibration between uncertainty estimates and actual truthfulness. [**Models**] We conduct experiments with several strong open-source Instruct LLMs, including llama3-8B [21], Mistral-7B [23], Qwen2-7B [1], Qwen2-57B [2] and Falcon-7B [29]. [**Experimental Settings**] We use the semantic-structural trade-off parameter $\alpha = 0.5$ in all experiments in Table 1. The sentence embedding model $\mathcal{M}_{\text{sentence}}$ used in Eqn 4.2 is mpnet-base-v2 [28]. We employ the POT library [7] to solve the $\text{D}_\alpha$ in Eqn 3.1. The $\mathcal{M}_{\text{entail}}$ used in Eqn 4.2 is Qwen2-32B-Instruct.

Table 1 demonstrates that GAUSS consistently yields the strongest negative correlation with factuality across models and datasets, validating its effectiveness in capturing uncertainty. In particular, on datasets like Bios and LongFact, GAUSS outperforms all baselines by a significant margin in terms of the SC and PC. Baselines such as Gen-Binary, LUQ, Dis-Rating, and Dis-Single exhibit weaker and inconsistent correlation, highlighting the limitations of simple LLM inferred uncertainty metrics. The Centrality method also yields weaker correlations, underscoring the need for paragraph-level modeling of structure and semantics in uncertainty estimation across generations. Beyond correlation, GAUSS also maintains strong calibration performance (UCCE/QCCE), often outperforming or closely matching the best among all methods. This balance between correlation and calibration confirms the value of incorporating structural and semantic alignment via graph-based paragraph representation.

### 5.1. Structural–Semantic Tradeoff

To assess the role of semantic and structural components in our uncertainty measure, we ablate the fusion weight $\alpha$ from 0 (semantic-only) to 1 (structural-only). As shown in Figure 4, both SC and PC correlations worsen at the extremes, with optimal performance consistently observed for intermediate $\alpha$. This confirms the complementary strengths of semantic similarity (capturing fact-level meaning) and structural alignment (capturing inter-fact relationships). Ignoring either leads to degraded performance, underscoring the importance of jointly modeling both for effective uncertainty

estimation in long-form generation.

## 6. Limitations and Future Work

While GAUSS provides an interpretable framework for long-form uncertainty quantification, several limitations remain:

- **Sources of uncertainty.** GAUSS captures overall generation variability without separating *epistemic* uncertainty from *aleatoric* uncertainty. Disentangling these sources is an important direction for future work.

- **Dependence on auxiliary models.** GAUSS relies on auxiliary models for atomic-fact extraction and entailment, so errors on subjective, rhetorical, or contested claims may propagate to the resulting graphs and uncertainty scores. This is especially relevant in ambiguity-heavy domains. Although Appendix P shows modest sensitivity to the decomposition model, GAUSS should be viewed as quantifying uncertainty over extracted factual content, rather than resolving nuanced or disputed claims.

## Conclusion

We introduce GAUSS, a structure- and semantics-aware framework for uncertainty quantification in long-form LLM generation. Unlike approaches that treat long-form text as a set of independent facts, GAUSS models both the semantic content of atomic units and their interdependencies within a paragraph to produce an uncertainty estimate.

## Impact Statement

This paper presents work whose goal is to advance uncertainty quantification in long form generations. There are many potential societal consequences of our work, none which we feel must be specifically highlighted here.

## Acknowledgment

This material is based upon work supported by the National Science Foundation under Grants No. 1956313 and No. 2334380.

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

# A. Proofs of Lemmas and Theorems

We provide the proofs of the Lemmas and Theorems below.

## A.1. Proof of Lemma 4.1

**Lemma 4.1** *(Lipschitz Continuity of Alignment Distance)* The alignment distance $D_\alpha(M^r, L^r)$ is Lipschitz continuous with respect to the feature cost matrix $M^r$ and the structural cost tensor $L^r$. That is, for any $(M^r, L^r)$ and $(\widetilde{M^r}, \widetilde{L^r})$,

$$\left| D_\alpha(M^r, L^r) - D_\alpha(\widetilde{M^r}, \widetilde{L^r}) \right| \leq (1-\alpha)\|M^r - \widetilde{M^r}\|_\infty + \alpha\|L^r - \widetilde{L^r}\|_\infty.$$

**Proof.** Recall the alignment distance is defined as:

$$D_\alpha(M^r, L^r) = \min_{\pi \in \Pi} \sum_{i,j,k,\ell} \left[(1-\alpha)M^r[i,j] + \alpha L^r[i,k,j,\ell]\right] \pi_{ij}\pi_{k\ell},$$

where $\Pi = \left\{ \pi \in [0,1]^{n_a \times n_r} \,\middle|\, \sum_{j=1}^{n_r} \pi_{ij} = 1, \sum_{i=1}^{n_a} \pi_{ij} = 1 \right\}$ is the set of admissible couplings between nodes of the anchor graph and reference graph with $n_a$ and $n_r$ nodes respectively.

For an arbitrary coupling $\pi \in \Pi$ we define:

$$D_\alpha(M^r, L^r; \pi) = \sum_{i,j,k,\ell} \left[(1-\alpha)M^r[i,j] + \alpha L^r[i,k,j,\ell]\right] \pi_{ij}\pi_{k\ell},$$

Suppose the cost inputs are perturbed as $M^r \mapsto \widetilde{M^r}$ and $L^r \mapsto \widetilde{L^r}$, and define

$$\Delta_M = \|M^r - \widetilde{M^r}\|_\infty, \quad \Delta_L = \|L^r - \widetilde{L^r}\|_\infty.$$

Let $\pi^*$ be an optimal coupling for $D_\alpha(M^r, L^r)$, and $\widetilde{\pi}^*$ be the optimal coupling for $D_\alpha(\widetilde{M^r}, \widetilde{L^r})$, then we observe that:

$$
\begin{aligned}
D_\alpha(M^r, L^r) - D_\alpha(\widetilde{M^r}, \widetilde{L^r}) &\leq D_\alpha(M^r, L^r; \widetilde{\pi}^*) - D_\alpha(\widetilde{M^r}, \widetilde{L^r}; \widetilde{\pi}^*) \\
&= \sum_{i,j,k,\ell} \left[(1-\alpha)(M^r[i,j] - \widetilde{M^r}[i,j]) + \alpha(L^r[i,k,j,\ell] - \widetilde{L^r}[i,k,j,\ell])\right] \widetilde{\pi}_{ij}^* \widetilde{\pi}_{k\ell}^* \\
&\leq \sum_{i,j,k,\ell} \left[(1-\alpha)\Delta_M + \alpha\Delta_L\right] \widetilde{\pi}_{ij}^* \widetilde{\pi}_{k\ell}^* \\
&= (1-\alpha)\Delta_M + \alpha\Delta_L,
\end{aligned}
$$

since $\sum_{i,j} \widetilde{\pi}_{ij}^* = \sum_{k,\ell} \widetilde{\pi}_{k\ell}^* = 1$.

By symmetry, the reverse inequality also holds. Hence,

$$\left| D_\alpha(M^r, L^r) - D_\alpha(\widetilde{M^r}, \widetilde{L^r}) \right| \leq (1-\alpha)\Delta_M + \alpha\Delta_L. \quad \blacksquare$$

## A.2. Proof of Theorem 4.1

**Theorem 4.1** *(Lipschitz Continuity of the Proposed Uncertainty Measure)*

Let $U(q)$ denote the uncertainty score for query $q$ with respect to a generated graph $G_a$ as the *anchor*, and $N-1$ independently generated graphs $\{G_r\}_{r \neq a}$ as *reference* graphs, defined as

$$U(q) = \frac{1}{N-1} \sum_{r \neq a} D_\alpha(M^r, L^r),$$

where $(M^r, L^r)$ are the semantic cost matrices and structural cost tensors between $G_a$ and $G_r$. Then $U(q)$ is Lipschitz continuous with respect to the collection $\{M^r, L^r\}_{r \neq a}$.

**Proof.** Recall that each alignment distance $D_\alpha(M^r, L^r)$ in $U(q)$ is Lipschitz continuous by **Lemma 4.1**:

$$\left| D_\alpha(M^r, L^r) - D_\alpha(\widetilde{M}^r, \widetilde{L}^r) \right| \leq (1 - \alpha) \|M^r - \widetilde{M}^r\|_\infty + \alpha \|L^r - \widetilde{L}^r\|_\infty.$$

Now define, for each $r$,

$$\Delta_M^r = \|M^r - \widetilde{M}^r\|_\infty, \qquad \Delta_L^r = \|L^r - \widetilde{L}^r\|_\infty.$$

Then

$$\begin{aligned}
\left| U(q) - \widetilde{U}(q) \right| &= \frac{1}{N-1} \left| \sum_{r \neq a} \left[ D_\alpha(M^r, L^r) - D_\alpha(\widetilde{M}^r, \widetilde{L}^r) \right] \right| \\
&\leq \frac{1}{N-1} \sum_{r \neq a} \left[ (1 - \alpha) \Delta_M^r + \alpha \Delta_L^r \right] \\
&\leq (1 - \alpha) \max_{r \neq a} \Delta_M^r + \alpha \max_{r \neq a} \Delta_L^r.
\end{aligned}$$

Since this bound depends linearly on the perturbations $\{M^r, L^r\}_{r \neq a}$, it shows that $U(q)$ is Lipschitz continuous in the collection $\{M^r, L^r\}$, as claimed.

### A.3. Proof of Theorem 4.2

**Theorem 4.2** *(Convergence of the Uncertainty Measure)* Under general boundedness assumptions on the alignment distances, the uncertainty score $U(q)$ defined exponentially converges around the true mean $E[U(q)]$, specifically:

$$\mathbb{P}\left[ |U(q) - \mathbb{E}[U(q)]| > \epsilon \right] \leq 2 \exp\left( -\frac{2(N-1)\epsilon^2}{D^2} \right),$$

for any $\epsilon$ and some constant $D > 0$ that depends on the graph generation inconsistency.

**Proof.** Let $G_a$ be the fixed anchor graph, and let $G_1, \ldots, G_{N-1}$ be the $N - 1$ reference graphs, each sampled independently from the LLM's output distribution. Recall

$$U(q) = \frac{1}{N-1} \sum_{r=1}^{N-1} D_\alpha(M^r, L^r),$$

where $D_\alpha(M^r, L^r) = D_\alpha(G_a, G_r)$ is the alignment distance between the anchor and the $r$-th reference. We further represent the uncertainty measure computed with graphs $\{G_1, \cdots, G_r, \cdots, G_{N-1}\}$ as $U(q; G_1, \ldots, G_r, \ldots, G_{N-1})$.

**Bounded differences.** Assume each term is bounded,

$$0 \leq D_\alpha(M^r, L^r) \leq D$$

for some constant $D > 0$. If we replace one reference graph $G_r$ by an independent draw $G_r'$, then only the $r$-th summand in $U(q)$ changes, and by boundedness,

$$\left| U(q; G_1, \ldots, G_r, \ldots, G_{N-1}) - U(q; G_1, \ldots, G_r', \ldots, G_{N-1}) \right| \leq \frac{D}{N-1}.$$

Hence $U(q)$ satisfies the bounded-differences property with constants $c_r = \frac{D}{N-1}$ for $r = 1, \ldots, N - 1$.

**Application of McDiarmid's Inequality.** By McDiarmid's inequality for any $\epsilon > 0$,

$$\Pr\left[ U(q) - \mathbb{E}[U(q)] > \epsilon \right] \leq \exp\left( -\frac{2\epsilon^2}{\sum_{r=1}^{N-1} c_r^2} \right) = \exp\left( -\frac{2(N-1)\epsilon^2}{D^2} \right).$$

A symmetric bound holds for $\Pr[\mathbb{E}[U(q)] - U(q) > \epsilon]$, so by the union bound,

$$\Pr\left[ |U(q) - \mathbb{E}[U(q)]| > \epsilon \right] \leq 2 \exp\left( -\frac{2(N-1)\epsilon^2}{D^2} \right),$$

as claimed.

**Remark 1:** Importantly, $D$ is a property of the underlying generative process, specifically, the language model's consistency in producing structurally and semantically similar paragraphs in response to the same query. Lower values of $D$ indicate that the model tends to generate paragraphs that are more coherent and uniform in their graph representations, thereby enabling faster convergence and more stable uncertainty estimates.

## B. `GAUSS-atomic` : Extending `GAUSS` to Atomic-Fact Uncertainty

---

**Algorithm 2** `GAUSS-atomic`

---

**Require:** LLM $\mathcal{M}$, query $q$, atomic extractor $\mathcal{M}_{\text{atomic}}$, entailment model $\mathcal{M}_{\text{entail}}$, sample count $N$

1: Sample $\{P_1, \ldots, P_N\} \sim \mathcal{M}(q)$
2: Anchor $P_a \leftarrow P_1$
3: Extract anchor facts $\{v_i\}_{i=1}^{n_a} \leftarrow \mathcal{M}_{\text{atomic}}(P_a)$
4: **for** $r = 2, \ldots, N$ **do**
5:     Extract semantic graph of $P_r$ and compute

$$\left(M^r, \pi^{r^*}\right) \xleftarrow{\text{GAUSS}} (G_a, G_r),$$

6: **end for**
7: **for** $i = 1, \ldots, n_a$ **do**
8:

$$U_{\text{fact}}(i) \leftarrow \frac{1}{N-1} \sum_{r=2}^{N} \cdot \frac{1}{n_r} \sum_{j=1}^{n_r} (1 - \pi_{ij}^{r^*})\, M^r[i,j]$$

9: **end for**
10: Compute calibration metrics (AUROC, AUARC, etc) on $\{U_{\text{fact}}(i)\}$

---

While `GAUSS` produces a single uncertainty score at the paragraph level, its design naturally lends itself to fine-grained extensions. We propose `GAUSS-atomic`, a fact-level variant of `GAUSS`, which "zooms in" on each atomic fact within the anchor paragraph and quantifies its uncertainty by evaluating alignment costs against reference generations. Specifically, `GAUSS-atomic` assigns an uncertainty score to each atomic fact by jointly considering its structural role and semantic correspondence within the broader context of the paragraph, thereby preserving the interpretability and rigor of the original `GAUSS` framework at a finer resolution.

Concretely, given a prompt $q$ and an LLM $\mathcal{M}$, we:

1. Draw $N$ independent long–form samples $\{P_1, \ldots, P_N\} \sim \mathcal{M}(q)$.

2. Designate $P_a = P_1$ as the *anchor*, and extract its atomic facts

$$\{f_i\}_{i=1}^{n_a} = \mathcal{M}_{\text{atomic}}(P_a).$$

3. For each reference $P_r$ $(r = 2, \ldots, N)$:
   - Extract its semantic graph and compute the $D_\alpha$ -alignment coupling $\pi^{r^*} \in \mathbb{R}^{n_a \times n_r}$ and cost matrix $M^r[i,j]$ between anchor nodes $v_i$ and reference nodes $w_j$.

4. Finally, for each anchor fact $v_i$ we define its atomic-fact uncertainty

$$U_{\text{fact}}(i) = \frac{1}{N-1} \sum_{r=2}^{N} \cdot \frac{1}{n_r} \sum_{j=1}^{n_r} (1 - \pi_{ij}^{r^*})\, M^r[i,j]$$

5. These $\{U_{\text{fact}}(i)\}$ can then be evaluated against ground-truth veracity labels to obtain calibration metrics like the AUROC, AUARC etc.

We provide the algorithmic flow for `GAUSS-atomic` in Algorithm 2. We illustrate the benefits of using `GAUSS` over other approaches in Table 2.

*Table 2.* Atomic calibration performance (AUROC ↑, AUARC ↑) on LongFact and WildHallu for four methods across five LLMs.

| Method / Model | LongFact | | WildHallu | |
|---|---|---|---|---|
| | AUROC | AUARC | AUROC | AUARC |
| **falcon-7b-instruct** | | | | |
| GEN-BINARY | 0.8462 | 0.9451 | 0.7859 | 0.8753 |
| DIS-RATING | 0.5143 | 0.8676 | 0.5032 | 0.7337 |
| DIS-SINGLE | 0.5000 | 0.8304 | 0.5001 | 0.7358 |
| `GAUSS-atomic` | **0.8600** | **0.9575** | **0.7853** | **0.8757** |
| **llama3-8b-instruct** | | | | |
| GEN-BINARY | 0.7205 | 0.9606 | 0.7876 | 0.9378 |
| DIS-RATING | 0.6839 | 0.9557 | 0.7479 | 0.9310 |
| DIS-SINGLE | 0.7560 | 0.9680 | **0.7984** | **0.9506** |
| `GAUSS-atomic` | **0.7616** | **0.9687** | 0.7847 | 0.9395 |
| **qwen2-7b-instruct** | | | | |
| GEN-BINARY | 0.7449 | 0.9587 | 0.7818 | 0.9298 |
| DIS-RATING | 0.6606 | 0.9404 | 0.6976 | 0.9025 |
| DIS-SINGLE | 0.7709 | 0.9657 | 0.7821 | 0.9334 |
| `GAUSS-atomic` | **0.7917** | **0.9674** | **0.7855** | **0.9382** |
| **qwen2-57b-instruct** | | | | |
| GEN-BINARY | 0.7192 | 0.9696 | 0.7762 | 0.9415 |
| DIS-RATING | 0.7530 | 0.9689 | 0.7494 | 0.9352 |
| DIS-SINGLE | 0.7830 | 0.9781 | 0.7685 | 0.9421 |
| `GAUSS-atomic` | **0.7920** | **0.9776** | **0.7691** | **0.9424** |
| **mistral-7b-instruct** | | | | |
| GEN-BINARY | 0.7372 | 0.9659 | 0.7556 | 0.9282 |
| DIS-RATING | 0.6821 | 0.9551 | 0.6674 | 0.8940 |
| DIS-SINGLE | 0.5472 | 0.9163 | 0.6065 | 0.8539 |
| `GAUSS-atomic` | **0.7612** | **0.9688** | **0.7689** | **0.9301** |

## C. Filtering Atomic Facts via `GAUSS-atomic`

One application of the `GAUSS-atomic` uncertainty scores is to improve the overall reliability of the generated content by *filtering out* the most uncertain atomic facts. Concretely, given a generated paragraph $P$, we:

1. Decompose $P$ into $n_a$ atomic facts $\{f_i\}_{i=1}^{n_a}$.

2. For each fact $f_i$:

   - Compute its uncertainty score $u_i = U_{\text{fact}}(i)$ via `GAUSS-atomic`.
   - Obtain its binary veracity label $s_i \in \{0, 1\}$ via the SAFE fact-checking module.

3. Choose a filtering level $k\%$: discard the top $k\%$ most-uncertain facts.

4. Let $\tau = \text{percentile}(\{u_i\}, 100 - k)$. Keep only $\mathcal{S} = \{i : u_i \leq \tau\}$, and compute the *filtered mean veracity*

$$\bar{v}_{\text{filtered}} = \frac{1}{|\mathcal{S}|} \sum_{i \in \mathcal{S}} s_i.$$

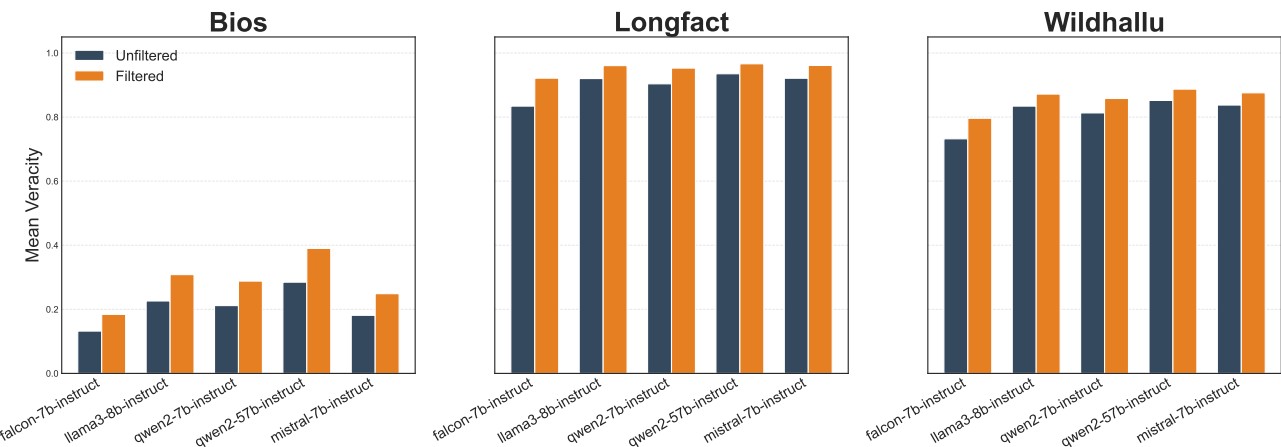

*Figure 5.* The increase in mean veracity $\bar{v}_{\text{filtered}}$ of the uncertainty based filtering - over the unfiltered-case $\bar{v}_{\text{all}}$

5. Compare $\bar{v}_{\text{filtered}}$ against the unfiltered mean $\bar{v}_{\text{all}} = \frac{1}{n_a} \sum_i s_i$ to measure the gain in factuality by uncertainty based filtering.

---

**Algorithm 3** Atomic-Fact Filtering via `GAUSS-atomic`

---

**Require:** Paragraph $P$, filtering rate $k \in [0, 100]$
1: $\{f_i\} \leftarrow$ `decompose_into_atomic_facts`$(P)$
2: $n_a \leftarrow |\{f_i\}|$
3: **for** $i = 1$ to $n_a$ **do**
4:     $u_i \leftarrow$ `GAUSS-atomic`$(f_i)$
5:     $s_i \leftarrow$ `SAFE_veracity`$(f_i)$
6: **end for**
7: $\bar{v}_{\text{all}} \leftarrow \frac{1}{n_a} \sum_{i=1}^{n_a} s_i$
8: $\tau \leftarrow \text{percentile}(\{u_i\}, 100 - k)$
9: $\mathcal{S} \leftarrow \{i : u_i \leq \tau\}$
10: $\bar{v}_{\text{filtered}} \leftarrow \frac{1}{|\mathcal{S}|} \sum_{i \in \mathcal{S}} s_i$
11: $\left(\bar{v}_{\text{all}}, \bar{v}_{\text{filtered}}\right)$

---

This method provides a simple, yet practical guide to obtain filtered atomic facts that have a strong guarantee to have higher veracity. We provide the algorithmic flow of the uncertainty based filtering approach in Algorithm 3 and the results of this approach in Figure 5.

## D. Alternative Notions of Structural Cost

Beyond simple pairwise local distances, we can capture richer, global connectivity in our semantic graphs by using the *heat kernel* of the graph Laplacian. Concretely, given a semantic graph

$$G_i = (V_i, E_i, C_i, \ell_f)$$

with $n_i$ nodes, let $L_i \in \mathbb{R}^{n_i \times n_i}$ be its (normalized) graph Laplacian. The heat kernel at time $\tau > 0$ is defined as

$$H_i(\tau) = \exp\left(-\tau L_i\right) \in \mathbb{R}^{n_i \times n_i}, \tag{D.1}$$

where $\exp$ denotes the matrix exponential. Entries $H_i(\tau)_{k,\ell}$ encode the flux from node $k$ to node $\ell$ over time $\tau$, thereby reflecting both local and multi-hop structural relationships.

To incorporate this into our alignment distance, we replace the original structural cost tensor $L^r$ (based on raw distance or cosine-dissimilarity) by a *heat-kernel cost tensor* $\widehat{L}^r(\tau)$ defined as

$$\widehat{L}^r(\tau)\big[i,j,k,\ell\big] \;=\; \big|\,H_a(\tau)_{i,k} \;-\; H_r(\tau)_{j,\ell}\,\big|. \tag{D.2}$$

Here $H_a(\tau)$ and $H_r(\tau)$ are the heat kernels of the anchor and reference graphs, respectively. We utilize $\tau = 0.99$ for the results shown in Table 3.

Finally, the full alignment distance between $G_a$ and $G_r$ becomes

$$\mathrm{D}^{\mathrm{heat}}_\alpha\big(G_a, G_r; \tau\big) = \min_{\pi \in \Pi} \sum_{i,j,k,\ell} \Big[(1-\alpha)\,M^r_{i,j} \;+\; \alpha\,\widehat{L}^r_{i,j,k,\ell}(\tau)\Big]\,\pi_{ij}\,\pi_{k\ell}\,, \tag{D.3}$$

which we then plug into the uncertainty measure $U(q)$.

*Table 3.* Comparison of `GAUSS-kernel` ($\tau = 0.99$) and `GAUSS` on atomic-calibration metrics (SC, PC, UCCE, QCCE) for LongFact and WildHallu (lower is better).

| Model / Method | LongFact | | | | WildHallu | | | |
|---|---|---|---|---|---|---|---|---|
| | SC | PC | UCCE | QCCE | SC | PC | UCCE | QCCE |
| **falcon-7b-instruct** | | | | | | | | |
| `GAUSS-kernel` | -0.6428 | -0.6767 | 0.2089 | 0.2777 | -0.7624 | -0.7682 | 0.1809 | 0.2078 |
| `GAUSS` | -0.6555 | -0.6915 | 0.1817 | 0.2199 | -0.7565 | -0.7616 | 0.1470 | 0.1978 |
| **llama3-8b-instruct** | | | | | | | | |
| `GAUSS-kernel` | -0.4038 | -0.4224 | 0.1702 | 0.2515 | -0.6789 | -0.7080 | 0.2133 | 0.2439 |
| `GAUSS` | -0.4433 | -0.4505 | 0.1613 | 0.2535 | -0.6808 | -0.7144 | 0.2048 | 0.2624 |
| **qwen2-7b-instruct** | | | | | | | | |
| `GAUSS-kernel` | -0.4891 | -0.5155 | 0.1522 | 0.2869 | -0.6999 | -0.7017 | 0.1799 | 0.2378 |
| `GAUSS` | -0.4979 | -0.5233 | 0.1403 | 0.2362 | -0.7072 | -0.7154 | 0.1798 | 0.2212 |
| **qwen2-57b-instruct** | | | | | | | | |
| `GAUSS-kernel` | -0.3582 | -0.4006 | 0.2367 | 0.3502 | -0.6644 | -0.7190 | 0.1966 | 0.2664 |
| `GAUSS` | -0.4226 | -0.4615 | 0.1844 | 0.3111 | -0.6852 | -0.7032 | 0.2061 | 0.2043 |
| **mistral-7b-instruct** | | | | | | | | |
| `GAUSS-kernel` | -0.4064 | -0.4491 | 0.2151 | 0.2815 | -0.6940 | -0.7591 | 0.1867 | 0.2243 |
| `GAUSS` | -0.4408 | -0.4678 | 0.2525 | 0.2672 | -0.6949 | -0.7584 | 0.1784 | 0.2310 |

We observe from Table 3 that employing straightforward semantic distance representations, such as adjacency matrices, to construct the structural cost tensor in `GAUSS` yields more effective calibration performance.

# E. Robustness to Semantic Embedding Variations

In order to explore the robustness of `GAUSS` to perturbations in the embeddings, we experiment with two different embedding models with the same embedding size. These models are `all-mpnet-base-v2` and `stsb-roberta-base`.

**`all-mpnet-base-v2`**   This model is fine-tuned on a broad mixture of semantic similarity datasets (e.g., STS, QuoraQP), and generates 768-dimensional embeddings. It is widely regarded as one of the strongest general-purpose sentence encoders in terms of semantic textual similarity (STS) performance.

**`stsb-roberta-base`**   This model builds on the RoBERTa-base encoder, fine-tuned on the STS Benchmark using a Siamese-BERT (SBERT) architecture. It also produces 768-dimensional embeddings that are optimized for capturing fine-grained relational semantics between sentence pairs.

Despite architectural and training differences, Table 4 reveals that both models yield comparable performance in calibration metrics across datasets. This empirical stability highlights a key theoretical property of our framework: as formalized

*Table 4.* Atomic-calibration metrics (SC, PC, UCCE, QCCE) for two embedding variants of GAUSS.

| Model / Embedding | Bios | | | | LongFact | | | | WildHallu | | | |
|---|---|---|---|---|---|---|---|---|---|---|---|---|
| | SC | PC | UCCE | QCCE | SC | PC | UCCE | QCCE | SC | PC | UCCE | QCCE |
| **falcon-7b-instruct** | | | | | | | | | | | | |
| GAUSS(all-mpnet-base-v2) | -0.4118 | -0.3321 | 0.1680 | 0.1845 | -0.6555 | -0.6915 | 0.1817 | 0.2199 | -0.7565 | -0.7616 | 0.1470 | 0.1978 |
| GAUSS(stsb-roberta-base) | -0.4080 | -0.3321 | 0.1832 | 0.1659 | -0.6727 | -0.7045 | 0.1974 | 0.2501 | -0.7517 | -0.7570 | 0.1829 | 0.2142 |
| **llama3-8b-instruct** | | | | | | | | | | | | |
| GAUSS(all-mpnet-base-v2) | -0.7066 | -0.7080 | 0.1035 | 0.1426 | -0.4433 | -0.4505 | 0.1613 | 0.2535 | -0.6808 | -0.7144 | 0.2048 | 0.2624 |
| GAUSS(stsb-roberta-base) | -0.7070 | -0.7045 | 0.1348 | 0.1035 | -0.5058 | -0.5240 | 0.1753 | 0.2559 | -0.6894 | -0.7141 | 0.2217 | 0.2707 |
| **qwen2-7b-instruct** | | | | | | | | | | | | |
| GAUSS(all-mpnet-base-v2) | -0.6915 | -0.7114 | 0.1606 | 0.1505 | -0.4979 | -0.5233 | 0.1403 | 0.2362 | -0.7072 | -0.7154 | 0.1798 | 0.2212 |
| GAUSS(stsb-roberta-base) | -0.6953 | -0.7143 | 0.1420 | 0.1664 | -0.5618 | -0.5892 | 0.1623 | 0.2617 | -0.7108 | -0.7178 | 0.1808 | 0.2385 |
| **qwen2-57b-instruct** | | | | | | | | | | | | |
| GAUSS(all-mpnet-base-v2) | -0.6991 | -0.7018 | 0.1328 | 0.1092 | -0.4226 | -0.4615 | 0.1844 | 0.3111 | -0.6852 | -0.7032 | 0.2061 | 0.2043 |
| GAUSS(stsb-roberta-base) | -0.6928 | -0.6995 | 0.0746 | 0.0952 | -0.4636 | -0.5311 | 0.2055 | 0.3329 | -0.6658 | -0.7186 | 0.2038 | 0.2603 |
| **mistral-7b-instruct** | | | | | | | | | | | | |
| GAUSS(all-mpnet-base-v2) | -0.6643 | -0.6766 | 0.1443 | 0.1407 | -0.4408 | -0.4678 | 0.2525 | 0.2672 | -0.6949 | -0.7584 | 0.1784 | 0.2310 |
| GAUSS(stsb-roberta-base) | -0.6648 | -0.6663 | 0.1261 | 0.1453 | -0.4820 | -0.4754 | 0.2001 | 0.2819 | -0.6963 | -0.7596 | 0.1716 | 0.2250 |

in **Theorem 4.1**, the proposed uncertainty measure is Lipschitz continuous in its semantic cost inputs. Hence, bounded perturbations in node embeddings (e.g., due to switching between embedding models) induce only minor, controlled changes in uncertainty scores and thereby small changes in calibration. This result not only ensures robustness to sentence encoder variations but also validates the practical reliability of GAUSS across diverse embedding backbones.

## F. Additional Evaluation Datasets: ELI5 and SciQA

*Table 5.* Calibration metrics (SC, PC, UCCE, QCCE) on ELI5 and SciQA.

| Model / Method | ELI5 | | | | SciQA | | | |
|---|---|---|---|---|---|---|---|---|
| | SC | PC | UCCE | QCCE | SC | PC | UCCE | QCCE |
| **falcon-7b-instruct** | | | | | | | | |
| DIS-RATING | 0.1079 | 0.1261 | **0.0373** | 0.2185 | 0.0621 | 0.0118 | 0.0567 | 0.1140 |
| DIS-SINGLE | 0.0283 | -0.0335 | 0.1533 | 0.4962 | 0.0430 | 0.0707 | **0.0000** | 0.5490 |
| GEN-BINARY | -0.5586 | -0.5696 | 0.1470 | 0.1929 | -0.5054 | -0.5465 | 0.0828 | 0.1124 |
| LUQ | -0.3164 | -0.3051 | 0.1219 | **0.1622** | -0.1878 | -0.2671 | 0.0924 | **0.0950** |
| GAUSS | **-0.6022** | **-0.5704** | 0.1237 | 0.1671 | **-0.5503** | **-0.6021** | 0.0744 | 0.1568 |
| **mistral-7b-instruct** | | | | | | | | |
| DIS-RATING | -0.3568 | -0.3185 | **0.1195** | **0.1577** | -0.1706 | -0.1657 | 0.1318 | 0.1854 |
| DIS-SINGLE | 0.1122 | 0.1045 | 0.1402 | 0.1927 | -0.0096 | -0.0969 | 0.1537 | 0.1976 |
| GEN-BINARY | -0.3662 | -0.3186 | 0.1480 | 0.1653 | -0.3002 | **-0.4275** | 0.1832 | **0.0976** |
| LUQ | -0.1271 | -0.0755 | 0.1761 | 0.1880 | -0.1071 | 0.0317 | 0.1372 | 0.1659 |
| GAUSS | **-0.3775** | **-0.3214** | 0.1699 | 0.2242 | **-0.3102** | -0.4220 | **0.1177** | 0.0995 |

To further assess our uncertainty quantification framework, we experiment on two publicly available long-form QA benchmarks:

**ELI5** The ELI5 dataset [6] is drawn from the "Explain Like I'm Five" subreddit. It covers a broad range of user-curated explanatory queries, making it a challenging testbed for paragraph-level UQ.

**SciQA** SciQA [31] is a science-focused QA collection of questions sourced from elementary and middle-school science curricula. Its domain specificity and factual rigor stress-test our uncertainty estimates in technical contexts.

We observe from Table 5 that GAUSS produces consistently better correlation with the factuality values.

## G. Convergence of $U(q)$ with Sample Size and LLM Consistency

We empirically examine the convergence behavior of the uncertainty measure $U(q)$ as a function of the number of reference samples $N$. According to Theorem 4.2, $U(q)$ concentrates around its expected value $\mathbb{E}[U(q)]$ at an exponential rate, with respect to the number of reference paragraphs $(N-1)$ and the graph generation inconsistency factor $D$ of the underlying LLM. As shown in Figure 6, the deviation $|U(q) - \mathbb{E}[U(q)]|$ diminishes as $N$ increases. Convergence is also faster for some models over others.

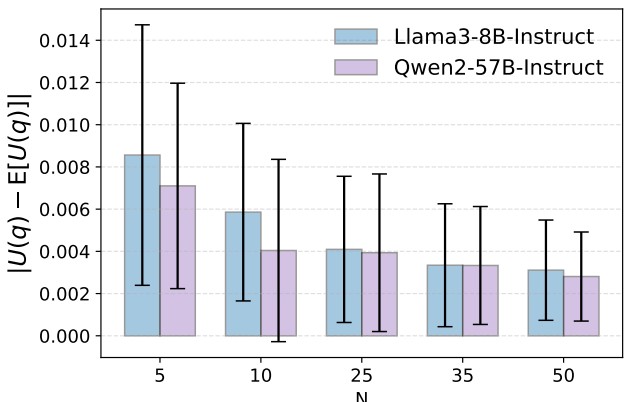

*Figure 6.* Convergence of the uncertainty measure $U(q)$ to its expected value as the number of reference samples $N$ increases.

To further quantify this behavior, we interpret the parameter $D$ as an intrinsic property of an LLM that governs the rate at which its empirical uncertainty estimates converge to the true expected uncertainty. For a fixed failure probability i.e. $\mathbb{P}[\,|U(q) - \mathbb{E}[U(q)]| > \epsilon\,] = \delta$, sample size $N$, and tolerance $\epsilon$, the high-probability upper bound on $D$ is given by:

$$D = \epsilon \sqrt{\frac{2(N-1)}{\ln(2/\delta)}}.$$

By setting $\delta = 0.05$ and averaging over multiple runs, we compute the mean upper bound of $D$ for two representative models:

$$D_{\texttt{Qwen-2-57B-Instruct}} \approx 0.0129, \quad D_{\texttt{LLaMA3-8B-Instruct}} \approx 0.0149.$$

Lower values of $D$ indicate LLMs with reduced graph inconsistency and faster convergence of their uncertainty estimates. This formulation offers a model-agnostic metric to compare LLMs based on their consistency in generating structurally and semantically coherent outputs across sampled generations.

## H. Choice of Anchor Paragraph

Our uncertainty estimation procedure requires selecting an anchor paragraph $G_a$ to compare against the remaining generations for a given query. By default, we follow prior work [34, 35], which uses the first generated paragraph as the anchor. While simple and reproducible, this choice may introduce variability in the resulting uncertainty scores. We thereby perform experiments with different anchor paragraphs below.

**Empirical Stability on Bios / `llama3-8b-instruct`**

To evaluate the robustness of our setup to anchor choice, we conduct two empirical analyses on the Bios dataset.

**1. Coefficient of Variation.** For each query, we compute the uncertainty score $U(q)$ using every paragraph in the set as the anchor, and record the coefficient of variation:

$$\text{CV} = \frac{\sigma(U(q))}{\mu(U(q))}.$$

where $\mu(\cdot), \sigma(\cdot)$ are the empirical mean and standard deviation across different anchor paragraphs for a given query. Averaged across queries, we find $\overline{\text{CV}} = 0.2333$, suggesting relatively low spread in uncertainty estimates.

**2. Correlation-Coefficient Variability.** We also assess how anchor choice affects agreement with ground-truth veracity labels by computing the variability in correlation metrics across anchors:

| Metric | Mean | Std Dev |
|--------|------|---------|
| SC | $-0.6844$ | $0.3799$ |
| PC | $-0.6826$ | $0.4077$ |

Taken together, these analyses indicate that while the uncertainty scores exhibit moderate stability across anchor choices, the choice of anchor is not entirely inconsequential, particularly when evaluating correlation with factuality at the dataset level. We therefore investigate a more principled anchor selection mechanism. Rather than using the first generated paragraph, we select a representative paragraph based on its semantic graph: the graph medoid, defined as the generated paragraph whose graph has the smallest average alignment distance to all other generated graphs. This choice reduces reliance on an arbitrary generation and mitigates the influence of atypical anchor samples. Importantly, the medoid is used only for anchor selection; the final uncertainty score continues to be computed using the original formulation of GAUSS.

Particularly, given the set of sampled paragraph graphs $\{G_i\}_{i=1}^N$ for a query, we select the anchor as the graph medoid under a lightweight pairwise fused Gromov–Wasserstein (FGW) distance:

$$G_a^{\text{medoid}} = \arg\min_{G_i} \sum_{j \neq i} D_\alpha^{\text{med}}(G_i, G_j). \tag{H.1}$$

Here, $D_\alpha^{\text{med}}(G_i, G_j)$ follows the same semantic–structural FGW form as the GAUSS alignment distance, but uses a simplified semantic cost to avoid invoking the entailment gate during anchor selection. Specifically, for a candidate pair of graphs $(G_i, G_j)$, we define

$$M_{\text{med}}^{(j)}[u, v] = 1 - \cos(\ell_f(u), \ell_f(v)), \tag{H.2}$$

where $u \in V_i$ and $v \in V_j$ denote atomic-fact nodes from the two candidate graphs. In contrast to the semantic cost used in the final GAUSS uncertainty computation in Eq. (4.2), this lightweight cost omits the entailment-based gating term. We retain the same structural discrepancy term as in the original framework:

$$L^{(j)}[u, u', v, v'] = |C_i(u, u') - C_j(v, v')|. \tag{H.3}$$

The resulting lightweight pairwise distance is

$$D_\alpha^{\text{med}}(G_i, G_j) = \min_{\pi \in \Pi} \sum_{u,v,u',v'} \Big[ (1 - \alpha) M_{\text{med}}^{(j)}[u, v] + \alpha L^{(j)}[u, u', v, v'] \Big] \pi_{uv} \pi_{u'v'}. \tag{H.4}$$

*Table 6.* Performance of GAUSS with graph-medoid anchor selection on two representative model–dataset pairs. Lower values are preferred for all metrics.

| Model | Dataset | SC ↓ | PC ↓ | UCCE ↓ | QCCE ↓ |
|-------|---------|------|------|--------|--------|
| qwen2-7b-instruct | LongFact | -0.4607 | -0.4959 | 0.1536 | 0.1823 |
| falcon-7b-instruct | WildHallu | -0.6804 | -0.6940 | 0.2214 | 0.2182 |

Importantly, this medoid-selection step is used only to choose the anchor graph. Once $G_a^{\text{medoid}}$ is identified, the final uncertainty score is computed using the original GAUSS distance, including the entailment-gated semantic cost defined in Eq. (4.2). Therefore, the additional anchor-selection procedure is entirely graph-based and does not require repeated entailment-model evaluations across candidate anchor pairs. Since it is performed once per query, the additional overhead remains practical.

Table 6 reports results for two representative model–dataset pairs using graph-medoid anchor selection. The medoid-based variant preserves meaningful negative correlations between uncertainty and factuality while providing a principled alternative to selecting the first generated paragraph as the anchor. These results indicate that GAUSS can accommodate a representative-anchor strategy without modifying its underlying uncertainty formulation.

Another lightweight strategy would be to select the paragraph with the median number of atomic fact nodes, which often approximates the central structure without requiring pairwise graph comparisons. We leave investigation into these and other anchor-selection strategies to future work.

## I. Computational Analysis of `GAUSS`

We analyze the computational cost and practical runtime of `GAUSS`. In practice, the runtime of `GAUSS` is dominated by three components: graph construction, entailment computation, and alignment.

Graph construction is efficient and uses lightweight embedding models such as `all-mpnet-base-v2`, which have modest parameter sizes and fast inference.

`GAUSS` performs graph alignment between the anchor paragraph and each of the $N - 1$ sampled reference paragraphs which involves the computation of the structural cost tensor. Overall the convex optimization process to find the final alignment distance between the anchor and all references has a complexity of

$$\mathcal{O}\left(N \cdot n_a^2 \cdot n_r^2\right),$$

where $N - 1$ is the number of references, $n_a$ is the number of atomic facts in the anchor, and $n_r$ is the number in each reference graph. Alignment is fully parallelizable and offloaded to an an AMD EPYC 7413 CPU.

Entailment, which is common to `GAUSS` and all other baselines, is performed using larger models like Qwen2-32B-Instruct and constitutes the primary computational bottleneck.

We compare `GAUSS` with Gen-Binary, which also depends on entailment checks, under a consistent hardware setup (Nvidia A100 GPUs for embedding and entailment). The runtimes across datasets are reported below:

| Dataset | # Queries | `GAUSS` Runtime (mins) | Gen-Binary Runtime (mins) |
|---|---|---|---|
| WildHallu | 500 | $\approx 35$ | $\approx 30$ |
| LongFact | 500 | $\approx 35$ | $\approx 30$ |
| Bios | 183 | $\approx 10$ | $\approx 8$ |

Since the dominant cost for both methods lies in entailment inference, their runtimes are comparable. However, `GAUSS` yields stronger alignment with ground-truth veracity, achieving higher Spearman and Pearson correlations, making it more effective without added computational overhead.

## J. Extending `GAUSS` with Causal Dependency Modeling

`GAUSS` is designed to model within-paragraph relational structure among atomic facts. In its default instantiation, the structural matrix captures the semantic organization of facts within a paragraph:

$$C_i(p, k) = 1 - \cos(\ell_f(v_p), \ell_f(v_k)),$$

where semantically related facts are represented as structurally closer. The FGW structural term then evaluates whether this pattern of pairwise organization is preserved across generated paragraphs.

Importantly, the `GAUSS` framework is not restricted to this single notion of structure. For paragraphs involving explicit logical or causal flow, we augment the structural representation with directed causal dependencies between atomic facts. For each paragraph graph $G_i$, we construct a causal adjacency matrix $A_i \in \{0, 1\}^{n_i \times n_i}$, where

$$A_i(p, k) = \begin{cases} 1, & \text{if atomic fact } v_p \text{ causally leads to } v_k, \\ 0, & \text{otherwise.} \end{cases}$$

The matrix $A_i$ is inferred using a separate language model and is generally non-symmetric, allowing it to represent directed causal relationships. We then define the causal-augmented structural matrix as

$$C_i^{\text{causal}}(p, k) = [1 - \cos(\ell_f(v_p), \ell_f(v_k))] + [1 - A_i(p, k)].$$

*Table 7.* Performance of the causal-augmented variant of GAUSS across standard long-form factuality benchmarks. Lower values are preferred for all metrics.

| Model | Bios | | | | LongFact | | | | WildHallu | | | |
|---|---|---|---|---|---|---|---|---|---|---|---|---|
| | SC ↓ | PC ↓ | UCCE ↓ | QCCE ↓ | SC ↓ | PC ↓ | UCCE ↓ | QCCE ↓ | SC ↓ | PC ↓ | UCCE ↓ | QCCE ↓ |
| llama3-8b-instruct | −0.71 | −0.70 | 0.16 | 0.19 | −0.43 | −0.44 | 0.17 | 0.23 | −0.66 | −0.70 | 0.16 | 0.16 |
| qwen2-7b-instruct | −0.72 | −0.74 | 0.12 | 0.12 | −0.51 | −0.54 | 0.18 | 0.29 | −0.72 | −0.71 | 0.20 | 0.22 |
| falcon-7b-instruct | −0.42 | −0.35 | 0.18 | 0.20 | −0.64 | −0.66 | 0.24 | 0.23 | −0.74 | −0.75 | 0.14 | 0.11 |

This representation captures both semantic relatedness and directed causal organization among atomic facts. The corresponding structural cost tensor between the anchor graph $G_a$ and a reference graph $G_r$ is computed as

$$L^{r,\text{causal}}[i, j, k, \ell] = \left| C_a^{\text{causal}}(i, k) - C_r^{\text{causal}}(j, \ell) \right|.$$

Thus, FGW penalizes discrepancies not only in semantic co-organization, but also in the causal dependency structure expressed across generations.

**Evaluation on Standard Long-Form Factuality Benchmarks.** We evaluate this causal-augmented variant on the same three datasets used in the main experiments: Bios, LongFact, and WildHallu. We consider three representative generating models, namely llama3-8b-instruct, qwen2-7b-instruct, and falcon-7b-instruct. All remaining components of the evaluation protocol, including factuality estimation and the computation of SC, PC, UCCE, and QCCE, are kept unchanged from the main experiments.

Table 7 shows that the causal-augmented variant yields negative correlations between uncertainty and factuality across all model–dataset combinations. In particular, it achieves strong correlations for qwen2-7b-instruct on Bios and WildHallu, and for falcon-7b-instruct on LongFact and WildHallu. These results are broadly comparable to the default semantic-relational structural representation used in the main experiments. This demonstrates that GAUSS can incorporate richer notions of paragraph structure, such as directed causal dependencies, without changing its underlying graph-alignment-based uncertainty formulation.

## K. Additional Experimental Details

All experiments were conducted using NVIDIA A100 GPUs. For generating responses from large language models (LLMs), we employed a sampling strategy with a temperature of 1.0, top-p of 0.95, and a maximum token limit of 512. Each model was prompted to produce 20 distinct responses per input, aligning with the evaluation protocols for GAUSS LUQ, and GEN-BINARY. In computing the semantic cost matrix $M^r$, we binarize the semantic cost matrix by setting entries with values $\geq 0.6$ to 1 and the rest to 0, thereby emphasizing stronger semantic alignments. For the SAFE framework, relevant web pages were retrieved using the Serper API, a high-performance Google Search API known for delivering real-time search results with unparalleled speed.

## L. Prompt Design

We illustrate the prompt design for $\mathcal{M}_{\text{atomic}}$ in Table 6.

We also illustrate the prompting approach for the $\mathcal{M}_{\text{entail}}$ in Table 7.

Below is the prompt for the SAFE framework in Table 8.

## M. Limitations and Future Work

While GAUSS provides an interpretable framework for long-form uncertainty quantification, several limitations remain:

- **Sources of uncertainty.** GAUSS captures overall generation variability without separating *epistemic* uncertainty from *aleatoric* uncertainty. Disentangling these sources is an important direction for future work.

- **Dependence on auxiliary models.** GAUSS relies on auxiliary models for atomic-fact extraction and entailment, so errors on subjective, rhetorical, or contested claims may propagate to the resulting graphs and uncertainty scores. This is

*Table 8.* Prompt template for the $\mathcal{M}_{\text{atomic}}$ fact-decomposition model.

---

**$\mathcal{M}_{\text{atomic}}$ Prompt**

Please break down the following passage into independent fact pieces.

**Step 1:** For each sentence, split it into atomic facts, each containing exactly one subject–verb–object triple. If no explicit verb appears, use "be" as the predicate.

**Step 2:** Output every fact piece on its own line, prefixed with ### and without any extra formatting.

**Step 3:** Ensure each fact is fully self-contained: avoid pronouns (he, she, it, this, that), and always repeat the original noun.

**Examples:**
Michael Collins (born October 31, 1930) is a retired American astronaut and test pilot...
### Michael Collins was born on October 31, 1930.
### Michael Collins is retired.
...
• *League of Legends (often abbreviated as LoL) is a multiplayer online battle arena video game...*
### League of Legends is a video game.
...
• *Emory University has a strong athletics program, competing in the NCAA Division I ACC...*
### Emory University has a strong athletics program.
...

Now it's your turn. Here is the passage:

$\{\texttt{passage\_text}\}$

Return only the list of prefixed fact pieces.

---

*Table 9.* Prompt template for the $\mathcal{M}_{\text{entail}}$ support-checking model.

---

**$\mathcal{M}_{\text{entail}}$ Prompt**

Paragraph:
{paragraph}

Atomic Fact:
{fact}

Is the above atomic fact *supported* by the given paragraph?

Answer solely from the context—do *not* rely on external knowledge.
Do *not* provide explanations.

**Output:** Either yes or no.
**Answer:**

---

especially relevant in ambiguity-heavy domains. Although Appendix P shows modest sensitivity to the decomposition model, GAUSS should be viewed as quantifying uncertainty over extracted factual content, rather than resolving nuanced or disputed claims.

# N. Brier Score and Expected Calibration Error

We also report standard Brier score and expected calibration error (ECE) for the binarized veracity labels and the uncertainty score (which are typically defined for binary labels and probabilistic predictions). Our ground-truth veracity, however, is originally a continuous score $v(q) \in [0, 1]$ (fraction of supported atomic facts in the paragraph). To make it compatible with these binary calibration metrics, we first construct a binarized label

$$y(q) = \mathbf{1}\{v(q) > 0.5\},$$

treating paragraphs with factuality above $0.5$ as "high-factual" and the rest as "low-factual."

For each method $m$ and query $q$, we obtain a scalar uncertainty score $u_m(q)$ and interpret it as a probability after monotone rescaling to $[0, 1]$, with higher values corresponding to a higher probability of being incorrect (equivalently, lower probability of being correct). We then compute the *Brier score* for method $m$ in the standard way as the mean squared error between predicted probabilities and binary labels, and the *expected calibration error* (ECE) using 10 equal-width bins over $[0, 1]$: in each bin we compare the average predicted probability with the empirical error rate and aggregate the absolute differences weighted by the bin frequencies. The resulting Brier and ECE values for GAUSS and all baselines are reported in Table 9.

*Table 10.* Prompt template for the SAFE fact-checking model.

---

**SAFE Fact-Checking Prompt**

Your task is to fact-check the following statement.
This statement is extracted from a passage about a specific subject (e.g., a person, place, or event).

Assign a veracity label:
- 'S' if the statement is factually correct.
- 'NS' if the statement is factually incorrect.

For example, given that we have the statement and evidence as such, output the veracity label output and your brief analysis as such:
**Statement**: Lebron James is a basketball player.
**Evidence**: Lebron James is an American professional basketball player for the Los Angeles Lakers of the NBA.
**Analysis**: Lebron James is an American professional basketball player, so this is correct.
**Output**: S
Pay close attention to numbers, dates, and other details.

Now for the statement and evidence below, output your brief analysis and veracity label output in the above described format:

- **Statement:** {atomic_fact}

- **Evidence:** {retrieved_evidence}

- **Output:** ¡your output here¿

---

| Model | Dataset | Brier ↓ | | ECE ↓ | |
|---|---|---|---|---|---|
| | | GAUSS | Gen-Binary | GAUSS | Gen-Binary |
| Falcon-7B-Instruct | Bios | 0.2110 | 0.4544 | 0.3856 | 0.6503 |
| | LongFact | 0.0654 | 0.0870 | 0.1164 | 0.1747 |
| | WildHallu | 0.1193 | 0.1199 | 0.0981 | 0.1297 |
| LLaMA3-8B-Instruct | Bios | 0.2006 | 0.4243 | 0.3278 | 0.5787 |
| | LongFact | 0.0112 | 0.0309 | 0.0944 | 0.1497 |
| | WildHallu | 0.0603 | 0.0976 | 0.0797 | 0.1693 |
| Qwen2-57B-Instruct | Bios | 0.2339 | 0.6868 | 0.3442 | 0.7349 |
| | LongFact | 0.0147 | 0.0274 | 0.0912 | 0.1137 |
| | WildHallu | 0.0645 | 0.0838 | 0.0425 | 0.1438 |

*Table 11.* Brier score and Expected Calibration Error (ECE) for GAUSS and Gen-Binary across models and datasets with binarized veracity labels. Lower is better.

## O. Practical Runtime of the GAUSS Distance Metric

Recall that the GAUSS uncertainty score for a query $q$ is defined as

$$U(q) = \frac{1}{N-1} \sum_{r \neq a} \mathrm{D}_\alpha(G_a, G_r),$$

where $G_a$ is the anchor graph and $\{G_r\}_{r \neq a}$ are the graphs induced by the remaining $N-1$ generations. Each term $\mathrm{D}_\alpha(G_a, G_r)$ in this summation is independent and can therefore be computed fully in parallel across references, so the overall uncertainty computation scales well with the number of sampled paragraphs. Section I of the appendix reports end-to-end runtimes for the full GAUSS pipeline, from atomic fact decomposition to final uncertainty scores.

The fused Gromov–Wasserstein distance $\mathrm{D}_\alpha(G_a, G_r)$ itself has a nominal complexity of $O(n_a^2 n_r^2)$ for graphs with $n_a$ and $n_r$ nodes, but in our setting $n_a, n_r \approx 50$ and the computation is fast in practice. This is largely because the underlying components of the cost, including the semantic term $M^r[i, j]$ and the structural term $C[i, k, j, \ell]$, are precomputed once per paragraph and reused across alignments. We report practical runtimes for computing $\mathrm{D}_\alpha(G_a, G_r)$ in Table 10 in the appendix.

| Operation | Setting | Mean Runtime |
|---|---|---|
| FGW distance $D_\alpha(G_a, G_r)$ | (per pair, averaged over all datasets) | 29.20 ms |
| Uncertainty $U(q)$ (GAUSS) | Bios, $N = 20$ generations | 22.544 s |
| Uncertainty $U(q)$ (GAUSS) | LongFact, $N = 20$ generations | 30.207 s |
| Uncertainty $U(q)$ (GAUSS) | WildHallu, $N = 20$ generations | 32.392 s |

*Table 12.* Practical runtimes for the fused Gromov–Wasserstein distance $D_\alpha(G_a, G_r)$ and the GAUSS uncertainty computation $U(q)$ with $N = 20$ generations.

## P. Atomic Fact Extraction: Noise Analysis

**Robustness to the choice of atomic fact extractor.** In most long-form uncertainty methods, depending on an auxiliary LLM to decompose paragraphs into atomic facts, there lies a concern on the atomic fact extraction noise. In our framework, we denote this decomposition model by $M_{\text{atomic}}$ (Qwen2-32B-Instruct in the main experiments) and the corresponding prompt by $p$ (given in Table 6 of the Appendix). For a given response paragraph, the atomic fact list produced by $(M_{\text{atomic}}, p)$ is denoted by $AL_{\text{orig}}$. We emphasize that this decomposition step is shared across GAUSS and fact-level baselines such as LUQ, Gen-Binary, and Centrality.

To analyse the extraction noise, we perform an ablation study that probes the sensitivity of GAUSS to variations in the atomic fact extraction process along two axes: (i) *changing the decomposition model* $M_{\text{atomic}}$ while keeping the prompt $p$ fixed, and (ii) *changing the prompt* while keeping $M_{\text{atomic}}$ fixed. Concretely, for (i) we vary the model size (Qwen2-7B, Qwen2-57B) under the same prompt $p$, and for (ii) we fix $M_{\text{atomic}}$ and consider two prompt variants: removing some facts in the few-shot examples from Table 6, and removing all few-shot examples entirely.

Given an alternative atomic fact list AL for the same paragraph, we quantify its similarity to $AL_{\text{orig}}$ using two complementary metrics:

1. **LLM-based overlap score.** For each atomic fact $f \in AL$, we query an LLM to decide whether $f$ is semantically present in $AL_{\text{orig}}$ (outputting 1 if supported and 0 otherwise). The overlap score between AL and $AL_{\text{orig}}$ is the mean of these binary labels over all $f \in AL$. We report the average overlap score across all responses in a dataset; higher values indicate that the two decompositions capture essentially the same factual content.

2. **GAUSS graph distance.** Each atomic fact list (both AL and $AL_{\text{orig}}$) induces a semantic graph under the GAUSS construction. We compute the GAUSS alignment distance $D(AL, AL_{\text{orig}})$ between the two induced graphs and report the mean distance across all responses; lower values indicate more similar graph structure and hence a more stable decomposition.

Empirically, we find that *changing the prompt* (while holding $M_{\text{atomic}}$ fixed) leads to noticeably *lower* overlap scores and *higher* graph distances, reflecting the fact that very different prompting can change how aggressively the model splits or merges facts. In contrast, when we *change the model size* (Qwen2-7B / 32B / 57B) under a fixed, well-specified prompt $p$, the overlap scores remain high and the GAUSS graph distances remain small, indicating that the resulting atomic fact lists are very close both semantically and structurally.

| Setting | $M_{\text{atomic}}$ | Prompt | Overlap $\uparrow$ | Mean Graph Dist. $\downarrow$ |
|---|---|---|---|---|
| Varying model, fixed $p$ | Qwen2-7B-Instruct | $p$ (orig.) | 0.9619 | 0.1288 |
| | Qwen2.5-57B-Instruct | $p$ (orig.) | 0.9823 | 0.0412 |
| Varying prompt, fixed Qwen2.5-32B | Qwen2.5-32B-Instruct | $p$ (few-shot pruned) | 0.9591 | 0.1786 |
| | Qwen2.5-32B-Instruct | $p$ (no few-shot) | 0.9037 | 0.1960 |

*Table 13.* Overlap score and GAUSS graph distance between alternative atomic fact extraction configurations and the reference decomposition ($M_{\text{atomic}}$ = Qwen2.5-32B-Instruct, prompt $p$). Higher overlap and lower distance indicate closer agreement with the reference atomic fact list.

These results suggest that, in practice, once a stable prompt $p$ is chosen (such as the one we provide in Table 6), the choice of $M_{\text{atomic}}$ has only a modest effect on the induced semantic graphs and the resulting GAUSS uncertainty scores. While GAUSS, like all fact-level methods, necessarily depends on an automatic decomposition step, this ablation indicates that a

carefully designed prompt largely controls the behavior of the extractor and prevents significant decomposition noise from dominating the uncertainty signal.

## Q. Comparison with traditional short-form UQ approaches

We compare GAUSS against two representative short-form uncertainty quantification methods designed for sentence or answer level generation: Semantic Uncertainty [17] and Generating with Confidence [18].

**Why short-form UQ does not directly extend to long-form settings.** Both approaches [17, 18] were originally developed for settings where each query is answered by *short* texts (typically a single sentence), and uncertainty is computed over sets of such short answers. Semantic Uncertainty clusters generations into semantic equivalence classes via NLI-based entailment tests and computes entropy over the resulting "semantic sets." Generating with Confidence constructs a semantic similarity graph over responses and defines dispersion metrics (e.g., degree-based and Laplacian-eigenvalue scores) on this graph.

In the long-form setting we consider, each generation is a *multi-sentence paragraph* with many entangled factual claims. Applying sentence-level NLI directly to full paragraphs is problematic for two reasons: (i) current NLI models are trained and evaluated primarily on sentence-scale premises and hypotheses and their performance degrades when inputs are long, noisy paragraphs; and (ii) NLI models explicitly lack access to the internal structure of atomic facts within a paragraph, which is precisely what GAUSS is designed to exploit.

**Our long-form adaptations of Semantic Uncertainty and Generating with Confidence.** To nonetheless provide a meaningful comparison, we implement *black-box, paragraph-level* variants of both methods that follow their original spirit but are computationally feasible for long-form outputs.

For Semantic Uncertainty [17], we treat each paragraph as a single response and approximate semantic equivalence between two responses via a semantic-similarity model (rather than exact NLI entailment). We then cluster multiple paragraphs into semantic sets and compute an entropy-based score over the empirical distribution of these sets, analogously to semantic entropy, and use this as the uncertainty measure for each query.

For Generating with Confidence [18], we construct a paragraph-level similarity graph where nodes are entire generations for a given query and edge weights are semantic similarities between paragraphs. On this graph we compute the degree-based uncertainty $U_{\text{Deg}}$ and eigenvalue-based uncertainty $U_{\text{EigV}}$ as defined in [18], and use them as baselines for long-form uncertainty.

| Model | Dataset | [18] Generating-With-Confidence | | | | [17] Semantic Entropy | | | |
|---|---|---|---|---|---|---|---|---|---|
| | | SC | PC | UCCE | QCCE | SC | PC | UCCE | QCCE |
| Falcon-7B-Instruct | Bios | -0.2946 | -0.2104 | 0.2584 | 0.2294 | -0.1535 | -0.1154 | 0.1863 | 0.1740 |
| | LongFact | -0.2888 | -0.3762 | 0.2448 | 0.2921 | -0.3258 | -0.3731 | 0.1652 | 0.2326 |
| | WildHallu | -0.5122 | -0.5491 | 0.1782 | 0.2227 | -0.5000 | -0.5194 | 0.1592 | 0.1305 |
| LLaMA3-8B-Instruct | Bios | -0.4715 | -0.4517 | 0.1244 | 0.1204 | -0.4307 | -0.4391 | 0.1601 | 0.2410 |
| | LongFact | -0.1355 | -0.2000 | 0.1692 | 0.2431 | -0.1446 | -0.2332 | 0.1574 | 0.2075 |
| | WildHallu | -0.5185 | -0.5162 | 0.1188 | 0.2578 | -0.4578 | -0.5567 | 0.1077 | 0.1933 |
| Qwen2-7B-Instruct | Bios | -0.4387 | -0.4497 | 0.1705 | 0.1739 | -0.3577 | -0.4702 | 0.0888 | 0.1106 |
| | LongFact | -0.2226 | -0.3704 | 0.1888 | 0.2523 | -0.1263 | -0.2825 | 0.0775 | 0.2370 |
| | WildHallu | -0.5978 | -0.6393 | 0.1680 | 0.2504 | -0.4749 | -0.5055 | 0.1666 | 0.2641 |
| Qwen2-57B-Instruct | Bios | -0.4454 | -0.4422 | 0.1171 | 0.1350 | -0.4979 | -0.4994 | 0.1336 | 0.2247 |
| | LongFact | -0.1845 | -0.1750 | 0.2578 | 0.2987 | -0.1231 | -0.1563 | 0.1728 | 0.2192 |
| | WildHallu | -0.5630 | -0.5758 | 0.1526 | 0.2650 | -0.4382 | -0.4722 | 0.1147 | 0.2161 |
| Mistral-7B-Instruct | Bios | -0.4185 | -0.4835 | 0.2045 | 0.1166 | -0.4540 | -0.4619 | 0.1489 | 0.1861 |
| | LongFact | -0.1782 | -0.3037 | 0.1911 | 0.2932 | -0.1644 | -0.2657 | 0.0791 | 0.1678 |
| | WildHallu | -0.5369 | -0.4962 | 0.1596 | 0.2250 | -0.4438 | -0.4944 | 0.1494 | 0.1846 |

*Table 14.* Short-form UQ baselines adapted to long-form: performance of Generating-with-Confidence [18] and Semantic Uncertainty [17], measured by Spearman (SC), Pearson (PC), and continuous calibration errors (UCCE/QCCE) across models and datasets.

**Results.** Table 13 reports Spearman correlation (SC), Pearson correlation (PC), and our continuous calibration metrics (UCCE/QCCE) for both baselines across models and datasets. Overall, we observe that short-form UQ methods, even when adapted to operate on long-form paragraphs, yield noticeably weaker correlations and poorer calibration than GAUSS (from the main text), supporting our hypothesis that modeling the internal semantic graph of atomic facts is important for long-form factuality and uncertainty.

