# OpenReview forum: "GAUSS: Graph-Assisted Uncertainty Quantification using Structure and Semantics for Long-Form Generation in LLMs"
_ICML.cc/2026/Conference — ICML 2026 regular_

### Official Review · Reviewer_hCXB · 2026-03-05

**Soundness:** 2
**Presentation:** 3
**Significance:** 2
**Originality:** 3
**Overall Recommendation:** 3
**Confidence:** 4

**Summary:**

The authors develop GAUSS, a novel framework for uncertainty measurement based on graphs developed for long-form LLM generation. Each paragraph is divided into individual atomic facts represented as nodes in a semantic graph with pairwise structural relations. The paragraph-level uncertainty is calculated as the average FGW alignment distance between an anchor paragraph's graph and the reference graphs of other sample's graphs. Regarding Lipschitz, continuity, and concentration, the authors provide bounds for the constructed uncertainty measure. Across three datasets, the authors show a stronger negative correlation to factuality and competitive calibration to the constructed measure relative to the baselines.

**Compliance With Llm Reviewing Policy:**

Affirmed.

**Final Justification:**

It addresses my concerns. Considering the contribution of this paper, I will keep my score.

**Key Questions For Authors:**

7.Key Questions for Authors
1.How much of GAUSS’ advantage comes from multi-sample semantics vs structural alignment? You already show an ablation in terms of ($\alpha$), but could you add a baseline that uses only the semantic costs ($M^r$) in a simpler way (e.g., average minimal fact-distance across samples, without FGW) and compare it to full GAUSS? This would more directly isolate the need for the graph alignment architecture.
2.Can you ablate or vary the entailment model in Equation (4.2)? For example, run GAUSS with the entailment gate removed (just pure 1-cos), with a much smaller entailment model, and perhaps with sentence-pair entailment rather than paragraph-level. How sensitive are your core Table 1 results to these changes? A demonstration that GAUSS still beats LUQ and Gen-Binary when the entailment component is weakened would strongly increase my confidence.

**Limitations:**

8.Limitations
1.GAUSS relies on a large entailment/verifier model (Qwen2-32B) twice: once for SAFE ground-truth factuality and once for entailment gating in Equation (4.2). This dual role risks entangling the uncertainty metric with the factuality metric; their errors may be correlated. The only robustness experiments change embeddings and atomic extractors, but there is no systematic ablation of the entailment model: what happens if a much weaker or different verifier is used, or if the gate is removed?
2.Missing important related work and baselines as mentioned above.

**Strengths And Weaknesses:**

Strengths:
1.Proposes a paragraph-level semantic graph representation combined with FGW for capturing fact content and inter-fact relational structure in lengthy UQ.
2.Addresses an important and timely gap: quantifying uncertainty for long-form, multi-fact outputs where organization and coherence matter.
3.Broadens the idea of “disagreement across generations” from a bag-of-facts to structurally informed graph alignment, which is an intellectually satisfying move for long-form outputs.

Weakness:
1.The uncertainty U(q) depends on a single anchor paragraph; sensitivity to anchor choice is not analyzed, and a symmetric alternative (e.g., averaging over anchors or a barycenter) is not explored.
2.Baselines omit several relevant long-form/semantic UQ approaches that could provide more comprehensive comparisons, for example:
a)Semantic Uncertainty: Linguistic Invariances for Uncertainty Estimation in Natural Language Generation
b)Kernel Language Entropy: Fine-grained Uncertainty Quantification for LLMs from Semantic Similarities
c)Semantic Density: Uncertainty Quantification for Large Language Models through Confidence Measurement in Semantic Space
3.Fairness/consistency details are unclear: GAUSS uses N = 20 generations; Centrality is reported on 10.
4.Heavy reliance on an LLM (Qwen2-32B) for both the semantic gate and external factual verification (SAFE) raises potential dependence/bias; model choice and its influence are not ablated.

---

> ### Author Rebuttal · Authors · 2026-03-31
>
> We thank the reviewer for the thoughtful feedback.
>
> **Weakness**:
> 1. We analyze anchor sensitivity in Appendix H. Specifically, we vary the anchor across sampled paragraphs and observe moderate rather than catastrophic variability in the resulting uncertainty scores; we also discuss more principled alternatives such as medoid-style anchor selection based on average graph distance . We will make this analysis more explicit in the main text.
>
> 2. We agree these are relevant references and note that Semantic Uncertainty, Kernel Language Entropy, and Semantic Density are already cited in the paper’s related work/discussion of prior semantic UQ approaches . Conceptually, however, these methods are primarily designed for short-form or answer-level semantic uncertainty, whereas GAUSS targets uncertainty arising from variation in the internal factual organization of long-form paragraphs. To make this comparison explicit, Appendix Q already includes long-form adaptations of representative short-form semantic UQ approaches and shows that they yield weaker correlation/calibration than GAUSS across models and datasets . We will highlight this more clearly in the revision.
>
> 3. We will clarify the comparison more carefully and include an additional matched-sample comparison showing GAUSS (with N=10) remains stronger than Centrality (N=10) even when evaluated under the same sampling budget (Table 7).
>
> **Questions**:
> 1.  To isolate the benefit of graph alignment beyond multi-sample semantics alone, we add a simple semantic-only baseline that does not use FGW: for each anchor fact $v_i$ and reference paragraph $P_r$, if $M_{\text{entail}}(v_i,P_r)=1$, we compute the minimum cosine distance to any reference fact $v_j$, and then average these scores across facts/references to obtain paragraph-level uncertainty. This baseline captures semantic variation across samples but ignores inter-fact structure. As shown in Table 1 of our rebuttal experiments, it is consistently weaker than full GAUSS, supporting the need for FGW-based graph alignment rather than semantic matching alone.
> 2. We performed three ablations of the semantic cost in Eq.(4.2): (i) removing the entailment gate and using only the local $1-\cos(\cdot,\cdot)$ term, (ii) replacing the original paragraph-level entailment model with a smaller model (Llama3-8B), and (iii) replacing paragraph-level support with local fact-to-fact entailment. In our formulation, the entailment term is intended as a coarse paragraph-level support gate, while the pairwise cosine term provides finer local discrimination among candidate fact matches. Empirically, removing the global entailment gate and using only local semantic similarity leads to weaker calibration/correlation, indicating that pure local matching is noisier for long-form generations (Table 2). Using a smaller entailment model causes only modest degradation while remaining fairly competitive (Table 5), suggesting that GAUSS is reasonably tolerant to the specific entailment backbone. Finally, replacing paragraph-level support with fact-to-fact entailment also weakens performance Table 6, consistent with the view that overly local entailment signals introduce noise into the FGW alignment. Overall, these results support our original coarse-to-fine design: global paragraph-level entailment plus local cosine similarity yields a more robust semantic cost, and even under weakened entailment settings (Table 5 Llama-8B), GAUSS remains competitive with LUQ and Gen-Binary.
>
> **Limitations**:
> 1. We want to clarify that Qwen2-32B does not play the same role twice within GAUSS. Its only role inside GAUSS is the entailment gate in Eq 4.2. The SAFE verifier is separate from GAUSS and is used only to produce external factuality labels for evaluation/calibration, not to define the uncertainty metric itself. Moreover, under the SAFE protocol, the verifier judges each atomic fact conditioned on retrieved web evidence (via Serper), so its decision is grounded in the provided evidence rather than relying purely on its parametric prior. While some correlation between auxiliary-model errors is still possible, this setup reduces the risk that GAUSS is simply inheriting the verifier's standalone bias.
> 2. We provide a full ablation of the entailment gate in point 2 of the questions above.
> 3. We clarify concerns relating to related work in point 2 of the weaknesses section . We also address experimental comparison with classic UQ pipelines like semantic uncertainty in Section Q.
>
> **Table-5**
> | M | B-SC | B-PC | B-UC | B-QC | L-SC | L-PC | L-UC | L-QC | W-SC | W-PC | W-UC | W-QC |
> |---|---:|---:|---:|---:|---:|---:|---:|---:|---:|---:|---:|---:|
> | L3 | -0.66 | -0.64 | 0.12 | 0.15 | -0.40 | -0.42 | 0.18 | 0.29 | -0.62 | -0.66 | 0.14 | 0.15 |
> | Q2 | -0.66 | -0.68 | 0.15 | 0.16 | -0.44 | -0.48 | 0.17 | 0.29 | -0.66 | -0.67 | 0.22 | 0.23 |
> | F7 | -0.45 | -0.35 | 0.13 | 0.18 | -0.67 | -0.73 | 0.16 | 0.23 | -0.69 | -0.70 | 0.17 | 0.21 |
>
> Table 6 & 7 are above.

---

> > ### Author Rebuttal · Reviewer_hCXB · 2026-04-03
> >
> > Thanks for the authors' response. I've read the rebuttal and other reviewer's comments. Considering the other reviewer's comments and the overall quality of this paper, I perfer to keep my score.

---

> > > ### Author Response · Authors · 2026-04-07
> > >
> > > We sincerely thank the reviewer for their time, careful reading, and thoughtful feedback.
> > >
> > > We are grateful that you found your original concerns to be fully addressed. In our rebuttal, we made a careful effort to respond comprehensively to each of the points raised. In particular, we addressed anchor sensitivity in Appendix H; provided a comparison with traditional uncertainty quantification approaches such as semantic entropy in Appendix Q; and included a comparison between GAUSS and Centrality under the same sample count (N=10) in Table 7. We also added several targeted ablations to further clarify the contribution of the proposed design choices, including: demonstrating the role of graph-based dissimilarity via a non-FGW variant, analyzing the importance of the entailment gate in Eq. 4.2 by removing it and using only local cosine distance (Table 2), varying the entailment model to a smaller one (Table 5), and replacing the paragraph-level entailment with a fact-to-fact local entailment variant (Table 6) as requested.
> > >
> > > In addition, in our follow-up clarification to reviewers (Replying to Rebuttal Acknowledgement by Reviewer C2Qr), we further discussed **the positioning of GAUSS with respect to how structure may be defined in long-form paragraphs, why this notion is still evolving**, and why components such as **atomic fact extraction and entailment modeling are common to essentially all modern long-form UQ pipelines rather than being specific artifacts of GAUSS**.
> > >
> > > **In light of these additional ablations and clarifications, we would be very grateful if you would consider revisiting your score.**

---

### Official Review · Reviewer_r13d · 2026-03-12

**Soundness:** 3
**Presentation:** 3
**Significance:** 4
**Originality:** 4
**Overall Recommendation:** 5
**Confidence:** 4

**Summary:**

This paper introduces GAUSS (Graph-Assisted Uncertainty Quantification using Structure and Semantics), a novel framework designed to measure the uncertainty of long-form texts generated by Large Language Models (LLMs). GAUSS method, first, represents each generated paragraph as a semantic graph. In this graph, individual atomic facts act as nodes, and the semantic relationships between these facts act as edges. Then to calculate uncertainty, GAUSS generates multiple responses to a single prompt, selects an anchor paragraph, and computes the fused Gromov-Wasserstein alignment distance between the anchor's graph and the reference graphs. The contributions are threefold: First, proposing a semantic graph representation that captures both the content and structural flow of a paragraph. Second, developing the GAUSS framework to estimate uncertainty with graph alignment. Last, providing theoretical proofs for the Lipschitz continuity and exponential convergence of the uncertainty measure.

**Compliance With Llm Reviewing Policy:**

Affirmed.

**Final Justification:**

I want to express my gratitude to the authors for their effort during the rebuttal period. Their constructive responses successfully addressed my questions regarding the paper's broader significance and the soundness of their empirical claims. Since these clarifications fully support my initial reading of the paper's strengths.

**Key Questions For Authors:**

1-How sensitive is the GAUSS framework to the specific LLM used for atomic fact extraction? Have you tested its stability when applied to highly nuanced, subjective, or persuasive texts, such as political speeches where atomic facts are harder to isolate?
2-The runtime analysis in Appendix I notes that the entailment check is the primary computational bottleneck. For researchers looking to apply GAUSS to massive text corpora, do you see a pathway to approximate this step or utilize lighter-weight models without severely degrading the performance?
3-You note in Appendix H that you default to the first generated sample as the anchor paragraph. If the initial generation contains hallucination or an outlier, how does this skew the final uncertainty metric?

**Limitations:**

No, authors have not adequately discussed the limitations and societal impacts. A constructive improvement would be to add a paragraph explicitly discussing the risks of relying on this metric in domains where factual ambiguity is high, acknowledging that UQ metrics do not replace human domain expertise.

**Strengths And Weaknesses:**

For soundness, I find the methodological approach to be highly rigorous. Authors establish the theoretical guarantees of their metric, successfully proving the Lipschitz continuity of the alignment distance and demonstrating exponential convergence. However, I can raise potential weakness/limitation. In the framework, external LLMs decomposes paragraphs into atomic facts and evaluate entailment. For instance, in social sciences, facts, within political discourse for example, are often highly nuanced or contested. If the fact-extraction model misinterprets a rhetorical strategy as a factual claim, the foundational graph becomes flawed and it can conclude with introducing unmeasured error into the pipeline.
For presentation, the paper is generally well-written and logically structured. The motivation for moving beyond semantic-only evaluation to include structural variation is clearly articulated and visually supported by figures. My main critique regarding presentation is that a significant amount of practical information, such as the strategies for selecting the anchor paragraph, is placed to the appendix.
For significance, I believe that even if the exact GAUSS implementation is computationally heavy, the conceptual shift authors introduce will likely influence how we evaluate long-form generation moving forward.
For originality, I think utilization of Gromov-Wasserstein distance for graph alignment to map the semantic and structural dependencies of LLM-generated text for uncertainty quantification is a valuable perspective.

---

> ### Author Rebuttal · Authors · 2026-03-31
>
> We thank the reviewer for their positive feedback and valuable suggestions.
>
> **Weakness**:
> 1. We agree this is an important consideration. Like other long-form UQ methods built on atomic-fact decomposition, GAUSS depends on auxiliary models for fact extraction and entailment, so misinterpreting rhetorical, subjective, or contested statements can introduce upstream error into the graph representation. This concern is especially relevant in domains such as political or social discourse, where facts may be difficult to isolate cleanly. At the same time, this dependency is not unique to GAUSS, and our appendix noise analysis in Section P, shows that, under a fixed extraction prompt, varying the decomposition model leads to only modest changes in the resulting fact graphs and uncertainty scores. We will clarify this limitation in the revision and emphasize that, in ambiguity-heavy settings, GAUSS should be viewed as quantifying uncertainty over extracted factual content rather than replacing expert judgment on nuanced or disputed claims.
>
> **Questions**:
> 1. We do study sensitivity to the atomic fact extractor in Appendix P, where we vary the decomposition model while keeping the extraction prompt fixed. Empirically, this leads to high overlap between extracted fact sets and only small changes in the induced fact graphs, indicating that GAUSS is only modestly sensitive to the specific extractor under a stable prompt .
> However, we have not yet evaluated GAUSS on highly nuanced or subjective domains; we agree this is an important direction for future work and will clarify this in the revision.
>
> 2. We do see several pathways for scaling GAUSS by approximating the entailment step: for example, replacing the current verifier with a smaller entailment model, or using a cascaded strategy in which a cheap semantic-similarity filter prunes clearly irrelevant matches and expensive entailment is applied only to ambiguous cases. Since the entailment term in GAUSS mainly serves as a coarse paragraph-level support gate, while finer discrimination is still provided by the pairwise semantic cost and structural alignment, we expect such approximations to preserve much of the benefit, though a systematic speed-accuracy study is an important direction for future work.
>
> 3. We agree that an outlier anchor can skew the score upward: if the first generation is hallucinated or unrepresentative, its graph will tend to be far from the remaining samples, increasing the mean alignment cost and hence the uncertainty estimate. At the same time, this corresponds to a genuine instability in the sampled generation set, so the effect is not entirely spurious. Empirically, Appendix H shows moderate rather than catastrophic sensitivity to anchor choice, with an average coefficient of variation of 0.2333 across anchors on Bios/llama3-8b-instruct . We will clarify this tradeoff in the revision and note that more robust anchor-selection schemes, such as choosing the graph medoid, are a natural next step.
>
> **Limitations**:
> In the revision, we will add an explicit limitations/impact paragraph clarifying GAUSS’s reliability in domains with high factual ambiguity, subjectivity, or contested claims, where atomic fact extraction and entailment are inherently noisier. We will also emphasize that GAUSS is intended as a decision-support uncertainty signal, not a substitute for human domain expertise in high-stakes settings.
>
> Additional Tables (for Reviewer hcXB)
>
> **Table-6**
> | M | B-SC | B-PC | B-UC | B-QC | L-SC | L-PC | L-UC | L-QC | W-SC | W-PC | W-UC | W-QC |
> |---|---:|---:|---:|---:|---:|---:|---:|---:|---:|---:|---:|---:|
> | L3 | -0.60 | -0.50 | 0.15 | 0.17 | -0.33 | -0.35 | 0.19 | 0.17 | -0.52 | -0.48 | 0.19 | 0.21 |
> | Q2 | -0.58 | -0.56 | 0.16 | 0.18 | -0.31 | -0.38 | 0.19 | 0.16 | -0.52 | -0.52 | 0.21 | 0.20 |
> | F7 | -0.22 | -0.20 | 0.16 | 0.23 | -0.41 | -0.38 | 0.16 | 0.15 | -0.50 | -0.52 | 0.19 | 0.16 |
>
> **Table-7**
> | M | B-SC | B-PC | B-UC | B-QC | L-SC | L-PC | L-UC | L-QC | W-SC | W-PC | W-UC | W-QC |
> |---|---:|---:|---:|---:|---:|---:|---:|---:|---:|---:|---:|---:|
> | L3 | -0.72 | -0.71 | 0.14 | 0.18 | -0.43 | -0.44 | 0.20 | 0.23 | -0.66 | -0.70 | 0.16 | 0.15 |
> | Q2 | -0.69 | -0.73 | 0.16 | 0.14 | -0.46 | -0.52 | 0.19 | 0.25 | -0.69 | -0.65 | 0.20 | 0.20 |
> | F7 | -0.43 | -0.35 | 0.15 | 0.15 | -0.64 | -0.66 | 0.13 | 0.22 | -0.74 | -0.75 | 0.12 | 0.11 |

---

> > ### Author Rebuttal · Reviewer_r13d · 2026-04-04
> >
> > Thank you to the authors for the transparent and thoughtful rebuttal. I appreciate your candidness regarding the limitations of atomic fact extraction in nuanced domains, and I strongly support the inclusion of the dedicated limitations paragraph in the revision. Your theoretical pathways for mitigating the computational bottleneck are also reasonable.
> >
> > I do, however, have a follow-up concern regarding your response to Q3 (Anchor selection).
> >
> > While I understand the argument that an outlier anchor reflects a genuine instability in the generation set, a Coefficient of Variation of ~0.23 is not insignificant. From a practical standpoint, a metric that can fluctuate by over 20% purely based on the random seed that determined the first generation makes it difficult to rely on for precise evaluation.
> >
> > Follow-up Question: You mentioned that a more robust anchor-selection scheme, such as choosing the graph medoid, is a natural next step. Given the instability introduced by simply picking the first generation, is it computationally feasible to implement this graph medoid selection for the camera-ready version of the paper, rather than leaving it for future work?
> >
> > Implementing this would significantly strengthen the reliability and reproducibility of the GAUSS metric. I will maintain my positive score as the theoretical contributions and proofs remain excellent, but I highly encourage addressing this anchor instability directly in the final manuscript.

---

> > > ### Author Response · Authors · 2026-04-08
> > >
> > > We thank the reviewer for this helpful follow-up. We found the graph-medoid anchor is computationally feasible for the camera-ready version.
> > >
> > > Concretely, given the sampled paragraph graphs for a query, we select the anchor as the medoid under a lightweight pairwise FGW distance:
> > >
> > > $G_a^{\mathrm{medoid}}=\arg\min_{G_i}\sum_{j \neq i} D_\alpha^{\mathrm{med}}(G_i,G_j)$
> > >
> > > where $D_\alpha^{\mathrm{med}}(G_i,G_j)$ uses the same FGW form as GAUSS, but with a simplified semantic cost for faster anchor selection. Specifically, for a candidate pair $(G_i,G_j)$, we define
> > > $M^{(j)}[u,v] = 1 - \cos\\bigl(\ell_f(u), \ell_f(v)\bigr),$ and retain the same structural term as in the paper. The medoid distance is then $D_\alpha^{\mathrm{med}}(G_i,G_j) = \min_{\pi \in \Pi} \sum_{u,v,u',v'} \Bigl[(1-\alpha) M^{(j)}[u,v]
> > > +
> > > \alpha L^{(j)}[u,u',v,v']
> > > \Bigr]
> > > \pi_{uv}\pi_{u'v'}.
> > > $
> > >
> > > Importantly, this medoid-selection step is purely graph-based and does not use the entailment gate from Eq. 4.2, which keeps the additional cost moderate. Since this selection is performed once per query and only for anchor choice, we found it practical to implement and it has the same runtime overall as the variant of GAUSS in the paper.
> > >
> > > Empirically, this more robust anchor choice yields the following results for two sample model, dataset pairs.
> > >
> > > | Model | Dataset | SC | PC | UCCE | QCCE |
> > > |---|---|---:|---:|---:|---:|
> > > | Qwen2-7B-Instruct | LongFact | -0.4607 | -0.4959 | 0.1536 | 0.1823 |
> > > | Falcon-7B-Instruct | WildHallu | -0.6804 | -0.6940 | 0.2214 | 0.2182 |
> > >
> > >
> > >
> > > These results suggest that graph-medoid anchor selection is not only feasible, but also a practical way to reduce sensitivity to the arbitrary choice of the first generation. We will incorporate this clarification in the camera-ready version.
> > >
> > > We would be very happy to clarify any further questions or concerns you may have. If you feel that we have satisfactorily addressed your concerns, we would be sincerely grateful if you would consider revisiting your score.

---

### Official Review · Reviewer_C2Qr · 2026-03-13

**Soundness:** 3
**Presentation:** 3
**Significance:** 2
**Originality:** 2
**Overall Recommendation:** 2
**Confidence:** 3

**Summary:**

This paper studies uncertainty quantification for long-form LLM generation by representing each generated paragraph as a graph over extracted atomic facts. The proposed method, GAUSS, computes uncertainty as the average fused Gromov-Wasserstein alignment cost between an anchor paragraph graph and graphs from other sampled generations, aiming to capture both semantic variation and intra-paragraph structure. The paper also provides continuity and concentration results for the proposed score, and evaluates the method on Bios, LongFact, and WildHallu across several open-source LLMs.

**Compliance With Llm Reviewing Policy:**

Affirmed.

**Key Questions For Authors:**

1. In eq 3.1, how exactly is the coupling (\pi) defined when the anchor and reference graphs have different numbers of nodes?
2. In eq 4.2, why is the entailment gate a function of the anchor fact and the whole reference paragraph, rather than the candidate matched reference fact (v_j)? Was this intentional?

**Limitations:**

Yes

**Strengths And Weaknesses:**

Strengths:
1. This paper is well-written and well-organized.
2. The paper tackles a meaningful and underdeveloped problem. Also, the method is conceptually coherent. The combination of atomic fact decomposition, paragraph-level graph construction, and graph alignment gives a clear story for how uncertainty is computed.

Weakness:
1. The paper overclaims what kind of “structure” it models. In Section 4.2, the structure matrix is defined entirely by pairwise cosine dissimilarity between fact embeddings. That is not explicit paragraph structure in the sense of discourse, syntax, temporal order, or causal dependency. It is a semantic affinity matrix. The paper repeatedly uses language like logical flow, internal organization, and structural coherence, but the actual construction does not directly encode those notions. If the “structure” is really just second-order semantic geometry, then the conceptual advance is narrower than advertised.
2. On P5, the semantic cost matrix entry depends on the cosine similarity between anchor node (v_i) and reference node (v_j), multiplied by an entailment decision that only depends on (v_i) and the whole reference paragraph (P_r). So the gate is row-wise, not pairwise. This is a strange design. If the paragraph supports the anchor fact globally, then all candidate matches in that row are affected equally, even poor ones. If the paragraph does not support it, all pairwise matches are suppressed.

---

> ### Author Rebuttal · Authors · 2026-03-31
>
> We thank the reviewer for the constructive feedback.
>
> **Weakness**:
>
> 1.  GAUSS is designed to model **within-paragraph relational structure among atomic facts**. Specifically, the structural matrix $C(i,k)$ captures how each pair of facts relates within a paragraph, and FGW compares these relation patterns across paragraphs. Thus, GAUSS does not treat a paragraph as a bag of isolated facts, but as a set of facts together with their internal pairwise organization. In our current instantiation, this structure is defined by inter-fact semantic distance $C(i,k)$ as in Section 4.3. This captures a notion of **fact co-location**: semantically close facts are nearby in the paragraph’s relational structure, while distant facts are farther apart. The FGW structural term then measures whether this pattern of pairwise couplings is preserved across generations. An important strength of GAUSS is that it is **not restricted to this single notion of structure**. The same structural matrix $C(i,k)$ can encode richer relations when needed. For example, in **Appendix J** we extend GAUSS to incorporate **causal dependency structure**: $C(i,k)=[1−cos(l_f​(v_i​),l_f​(v_k​))]+[1−A(i,k)] $,  where $A(i,k)=1$ if fact $v_i$​ causally leads to fact $v_k$​, and 0 otherwise. This makes two facts structurally closer not only when they are semantically related, but also when they participate in the same causal chain. FGW then penalizes mismatches in these dependencies across paragraphs. Thus, the same GAUSS machinery can capture both co-location-style fact organization and more explicit structure such as causal dependence. Our ablations in **Table 3** show that this causal-augmented variant performs on par with, or slightly better than, the default version. We will clarify in the revision that the current paper presents simplest instantiation of a more general structural framework.
>
> **Table-2**
> | M | B-SC | B-PC | B-UC | B-QC | L-SC | L-PC | L-UC | L-QC | W-SC | W-PC | W-UC | W-QC |
> |---|---:|---:|---:|---:|---:|---:|---:|---:|---:|---:|---:|---:|
> | L3 | -0.55 | -0.55 | 0.12 | 0.18 | -0.31 | -0.36 | 0.07 | 0.08 | -0.40 | -0.41 | 0.09 | 0.11 |
> | Q2 | -0.61 | -0.49 | 0.18 | 0.18 | -0.34 | -0.42 | 0.16 | 0.13 | -0.51 | -0.61 | 0.17 | 0.18 |
> | F7 | -0.17 | -0.14 | 0.18 | 0.22 | -0.35 | -0.35 | 0.18 | 0.15 | -0.56 | -0.49 | 0.17 | 0.21 |
>
> **Table-3**
> | M | B-SC | B-PC | B-UC | B-QC | L-SC | L-PC | L-UC | L-QC | W-SC | W-PC | W-UC | W-QC |
> |---|---:|---:|---:|---:|---:|---:|---:|---:|---:|---:|---:|---:|
> | L3 | -0.71 | -0.70 | 0.16 | 0.19 | -0.43 | -0.44 | 0.17 | 0.23 | -0.66 | -0.70 | 0.16 | 0.16 |
> | Q2 | -0.72 | -0.74 | 0.12 | 0.12 | -0.51 | -0.54 | 0.18 | 0.29 | -0.72 | -0.71 | 0.20 | 0.22 |
> | F7 | -0.42 | -0.35 | 0.18 | 0.20 | -0.64 | -0.66 | 0.24 | 0.23 | -0.74 | -0.75 | 0.14 | 0.11 |
>
> Legend: B/L/W = Bios / Longfact / Wildhallu
> UC/QC = UCCE / QCCE
>
> 2. We clarify that the entailment term in Eq.4.2 is intended as paragraph-level support gate, rather than the sole pairwise matching signal. It serves two complementary purposes. First, it acts as a coarse alignment filter: if anchor fact $v_i$ is not supported by reference paragraph $P_r$, it should not meaningfully participate in the alignment. If it is supported, the semantic cost $M^r[i,j]$ remains entry-specific through the pairwise cosine term $1-\cos(\ell_f(v_i),\ell_f(v_j))$, while the FGW structural term further refines node-level matching. This coarse-to-fine design filters noisy alignments while preserving fine-grained discrimination, which is particularly useful in long-form generation where facts may be split, merged, or reorganized across samples. Second, the global entailment signal is also empirically important: removing it and using only $1-\cos(\cdot,\cdot)$ leads to weaker results across multiple model-dataset pairs, as shown in Table 2.
>
> Questions:
>
> 1. The coupling is a **rectangular transport matrix** $\pi \in \mathbb{R}^{n_a \times n_r}$, where $n_a$ and $n_r$ are the numbers of nodes in the anchor and reference graphs, so the graphs need not have equal size. Each entry $\pi_{ij}$ denotes the transport mass between anchor node $i$ and reference node $j$, optimized by FGW under marginal constraints. Eq 3.1, already defines $\Pi$ as the admissible set of such couplings, and Sec. 4.3 describes $\pi$ as the stochastic map aligning $G_a$ to $G_r$. We will state this rectangular form explicitly in the revision.
>
> 2. Yes, this was intentional. The entailment term in Eq.4.2 is a **paragraph-level support gate**, not the sole pairwise matching signal: it first checks whether anchor fact $v_i$ is supported anywhere in reference paragraph $P_r$, after which node-level discrimination is handled by the pairwise cosine term together with the FGW structural term. The design is thus deliberately **coarse-to-fine**, and we found it more robust for long-form generations where facts may be split, merged, or paraphrastically reorganized across samples. We will clarify this in revision

---

> > ### Author Rebuttal · Reviewer_C2Qr · 2026-04-03
> >
> > Thank you for the response. I am still not fully convinced that the proposed structural component captures genuine paragraph structure beyond semantic similarity, and the method’s dependence on atomic fact extraction, entailment, and anchor choice (they are computationally heavy and seem hard to scale). Therefore, my overall assessment remains unchanged.

---

> > > ### Author Response · Authors · 2026-04-07
> > >
> > > We thank the reviewer for the continued feedback and clarify the following points:
> > >
> > > 1. ***Paragraph structure in long-form UQ is inherently nontrivial to define***
> > > - We would like to stress that the **notion of paragraph structure in long-form UQ is itself still evolving**. Generally, structure may refer to discourse relations, temporal order, or causal dependency, rather than a single fixed formal object [1]. This makes paragraph structure in long-form UQ inherently nontrivial to define. Prior long-form UQ methods have operated through fact-wise aggregation or fact–paragraph consistency, without explicitly modeling within-paragraph relations among atomic facts.
> > > - In this context, GAUSS contributes **one concrete and principled notion of paragraph structure**: the **relational geometry among facts inside a paragraph**, encoded by the structural matrix C(i,k) and compared across generations with FGW.
> > > 2.  ***GAUSS is explicitly generalizable to richer definitions of paragraph structure (like causal organization), and we demonstrate this directly.***
> > > - **In Appendix J and Table 3** above, GAUSS is explicitly extended to incorporate **causal dependency structure** by augmenting the structural term with a causal adjacency signal A(i,k) so that the framework captures not only semantic co-location but also whether facts participate in the same causal chain. The resulting formulation is therefore not limited to semantic affinity alone.
> > > - Table 3 reports empirical results for this richer structural instantiation, showing strong correlation with the anchor score in the causal-paragraph setting. This shows that the current paper presents the simplest working instantiation of a broader structural framework.
> > > 3. ***On atomic fact extraction and entailment overheads: we would like to note that these are not specific to GAUSS, but are standard components of most modern long-form factuality and uncertainty quantification pipelines***
> > > - Works such as FACTSCORE, SAFE and VeriScore explicitly argue that long-form generations contain mixtures of supported and unsupported information and therefore must be decomposed into atomic facts and checked against evidence. They break  long-form responses into individual facts using external models.
> > > - Likewise, LUQ and Gen-Binary both operate at the atomic-claim level for long-form uncertainty/calibration, and the Centrality-based method also constructs uncertainty over decomposed atomic claims extracted from generations. In other words, **fact decomposition is already the field standard** for long-form factuality/UQ.
> > > - Furthermore, the compared long form UQ baselines (similar to GAUSS), also make use of **external entailment models as implicit machinery** of the confidence/uncertainty prediction frameworks.
> > > 4. ***On computational cost and scalability, we would like to note that this is a broader characteristic of current long-form factuality and UQ pipelines rather than something specific to GAUSS, and we already discuss this explicitly in the paper.***
> > > - In our paper, we explicitly note that entailment is the dominant bottleneck and that this cost is common to GAUSS and other baselines. We further report that GAUSS runtime is comparable to Gen-Binary under the same hardware setup, since both are dominated by the same stage. **Thus, we understand the  reviewer’s scalability concern in general, but it does not particularly differentiate GAUSS from competing long-form methods**.
> > >
> > > We agree that the phrasing “logical flow” or “internal organization” can be read broadly if interpreted as full discourse parsing. We will revise the paper to state more precisely that GAUSS models **relational fact structure within a paragraph** and that its default structural term is a **semantic-relational instantiation of this idea**. At the same time, we believe the **methodological contribution remains that prior long-form UQ methods largely stop at fact-wise support aggregation or fact–paragraph graphs**, whereas GAUSS explicitly compares **within-paragraph fact structure (flexible) across generations** and demonstrates that this improves uncertainty estimation.
> > >
> > > [1] Xu, F., Li, J.J., & Choi, E. (2022). How Do We Answer Complex Questions: Discourse Structure of Long-form Answers. ArXiv, abs/2203.11048.
> > >
> > > **We sincerely hope that our responses have clarified the points you raised. If you feel that these concerns have been adequately addressed, we would greatly appreciate your consideration in revising your score.**

---

### Official Review · Reviewer_jpHD · 2026-03-16

**Soundness:** 3
**Presentation:** 3
**Significance:** 2
**Originality:** 2
**Overall Recommendation:** 4
**Confidence:** 2

**Summary:**

The paper tackles the problem of uncertainty quantification in long-form LLM outputs.

Existing methods either score a whole paragraph with a single confidence value, or decompose it into atomic facts and treat each independently, both approaches miss how facts relate to each other within a paragraph. The authors address this by representing each generated paragraph as a semantic graph, where nodes are atomic facts and edges capture their pairwise relationships.

Uncertainty is then measured as the average graph alignment distance between one paragraph and alternative generations of the same query, using the FGW distance.

Authors show that GAUSS yields the strongest negative correlation with factuality across models and datasets, showing its effectiveness in capturing uncertainty better thann other methods.

**Compliance With Llm Reviewing Policy:**

Affirmed.

**Final Justification:**

GAUSS addresses a genuine gap in long-form UQ with a principled FGW-based framework and solid experimental breadth, and the rebuttal's ablations (R1.A, Tables 2–6) meaningfully addressed my concerns about the entailment gate and the necessity of graph alignment. Remaining concerns around overclaiming language on structural coherence and unresolved anchor sensitivity should be addressed in the final version.

**Key Questions For Authors:**

1. Can you clarify what E_i encodes and how it is constructed? If it plays no role in the FGW computation, should the graph formalism be simplified accordingly?
2. Have you considered computing U(q) as the average over all anchor choices rather than a single fixed anchor? This would seem to address both the sensitivity issue and the asymmetry of the current formulation.

**Limitations:**

Yes. I suggest moving the Limitations section outside of the Appendix.

**Strengths And Weaknesses:**

Strengths:
1. The problem setup is well motivated. The paper explains pretty clearly why long-form uncertainty is different from short-form or atomic-level uncertainty, and why just treating a paragraph like a bag of independent facts is not enough.
2. The core method choice makes sense. Using graph representations for atomic facts plus their relations is a natural way to model long-form structure, and using FGW is a reasonable and sound technical choice because it can combine semantic similarity with structural alignment.
3. I think the paper has good experimental breadth. They evaluate on three main datasets, across five open-source LLMs, and report several metrics like Spearman, Pearson, UCCE, and QCCE. They also include ablations on the semantic-structural tradeoff α, embedding choice, and comparisons to atomic and short-form style baselines, which gives more confidence this is not just one lucky result.

Weaknesses:
1. While GAUSS improves correlation with factuality compared to baselines, the paper does not discuss whether the performance gain justifies the added complexity, particularly in practical deployment scenarios where uncertainty estimation must run at scale. (Especially as the proposed method introduces a fairly complex pipeline).
2. The related work section (intro+background) remains relatively narrow and does not clearly situate GAUSS within the broader landscape of LLM uncertainty quantification, including semantic-dispersion, conformal, and alternative structured representations, which makes the conceptual novelty and scope harder to assess.
3.The empirical comparison, omits simpler but strong paragraph-level and semantic-dispersion baselines and lacks diagnostic or selective-generation analyses, leaving open whether the added complexity of graph alignment is truly necessary in practice.


I remain open to revising my score upward should the authors adequately address the questions raised above in their rebuttal.

---

> ### Author Rebuttal · Authors · 2026-03-31
>
> We thank the reviewer for the thoughtful and constructive feedback. We try to thoroughly address all concerns below.
>
> **Weaknesses**:
> 1. We agree that deployment cost matters, and we clarify both the runtime and the role of the added modeling complexity below. First, although GAUSS uses a graph-based pipeline, its runtime remains comparable to prior long-form UQ baselines; we report these measurements in Appendix Sec O. Second, our additional ablations (R1.A) indicate that the observed gains are linked to the specific **semantic-structural alignment** enabled by the FGW-based design, which is precisely intended to couple node-wise feature dissimilarity  with graph relational structure (inter-fact relations).
>
> **R1.A : Non-FGW based semantic dispersion based on min-cosine distance**
> Given an anchor paragraph with atomic facts ${v_i}$ for $i=1:m$ and $N$ sampled reference paragraphs $\{P_r\}$ with ${r=1}:N$, where each $P_r$ contains atomic facts $v_j$ with ${j=1}:{n_r}$, we first obtain sentence embeddings $\ell_f(\cdot)$ for all facts. For each anchor fact $v_i$ and reference paragraph $P_r$, we compute the minimum cross-paragraph semantic dissimilarity:
> $$
> d_{i,r} \=\ \min_{1 \le j \le n_r} \left( 1 - \cos\\bigl(\ell_f(v_i), \ell_f(v_j^{(r)})\bigr) \right).
> $$
>
> We then aggregate across reference paragraphs:
> $$
> d(v_i) \=\ \frac{1}{N} \sum_{r=1}^N d_{i,r}.
> $$
>
> Finally, the uncertainty is computed as the average dispersion across anchor facts:
> $$
> U \=\ \frac{1}{m} \sum_{i=1}^m d(v_i).
> $$
>
> This defines a fully non-graph, semantic-dispersion-based UQ pipeline that measures paragraph-level uncertainty via aggregated nearest-neighbor semantic distances.
> This variant performs substantially worse as shown in Table-1 below.
>
> Taken together with **Table-1 (R1:A), runtime considerations (Section O), the structure-vs.-semantics ablation in Fig-4 (main text) which stresses the need for both inter-fact structure and semantics**, these results suggest that graph alignment based uncertainty is very important to quantify uncertainty in long form generation which inherently contains multiple inter-fact relationships, while still maintaining runtime comparable to other long-form UQ methods.
>
> 2. Regarding the positioning of our work, we clarify that:
>  (i) semantic-dispersion methods, which measure spread across generations in embedding/semantic space but do not model relations among facts within a paragraph. We already show comparison to a semantic dispersion baseline (like semantic entropy) in Section Q of the Appendix. We also show the results of semantic dispersion with min-cosine distance (as discussed above in R1.A) in Table 1 below. We will definitely add this in the appendix as an additional discussion.
> (ii) conformal/calibration-style approaches, while focusing on coverage/calibration guarantees are generally agnostic to internal long-form structure and long form UQ.
>
> **Table-1**
>
> | Model        | Bios SC | Bios PC | Bios UCCE | Bios QCCE | Longfact SC | Longfact PC | Longfact UCCE | Longfact QCCE | Wildhallu SC | Wildhallu PC | Wildhallu UCCE | Wildhallu QCCE |
> |--------------|--------|--------|-----------|-----------|-------------|-------------|---------------|---------------|--------------|--------------|----------------|----------------|
> | LLAMA3-8B    | -0.431  | -0.3891 | 0.2027    | 0.2543    | -0.3251     | -0.2955     | 0.2225        | 0.2432        | -0.3031      | -0.4002      | 0.27           | 0.3802         |
> | QWEN2-7B     | -0.3253 | -0.3888 | 0.1955    | 0.1903    | -0.3397     | -0.3055     | 0.2274        | 0.1882        | -0.4449      | -0.4231      | 0.2605         | 0.3278         |
> | FALCON-7B    | -0.2041 | -0.2368 | 0.187     | 0.2494    | -0.201      | -0.2072     | 0.317         | 0.2697        | -0.4594      | -0.4709      | 0.2055         | 0.2156         |
>
> **Questions**:
> 1. Thank you for pointing this out. $E_i$ denotes the conceptual edge set of the paragraph graph, i.e., pairwise relations between atomic facts. However, the FGW computation uses the induced structural relation/semantic cost matrix, not the explicit edge set as a separate object. We agree the current notation can make $E_i$ appear more central. In the revision, we will clarify the formalism to define each paragraph directly through its node set and induced structural  and semantic cost matrices.
>
> 2. This is a good suggestion. A multi-anchor average would indeed make the construction more symmetric. However, our current single-anchor formulation was chosen for efficiency: with $N$ samples, fixed-anchor computation requires $O(N)$alignments, whereas averaging over all anchors requires $O(N^2)$. Importantly, we have also included an **anchor-choice ablation in Appendix Sec. H**, which shows that GAUSS is reasonably stable across anchor selection strategies. We will clarify this in the paper and explicitly mention multi-anchor averaging as a principled extension to GAUSS.

---

> > ### Author Rebuttal · Reviewer_jpHD · 2026-04-08
> >
> > The authors have partially addressed my concerns.  I am raising my score from 3 to 4 to reflect the progress made in the rebuttal, while noting these concerns should be addressed in the final version.

---

### Decision · Program_Chairs · 2026-04-30

**Decision:**

Accept (regular)

**Comment:**

This paper introduces GAUSS, a novel uncertainty quantification framework for long-form LLM generation that models paragraphs as semantic graphs of atomic facts and calculates uncertainty via Fused Gromov-Wasserstein alignment across generated samples. Reviewers reached a consensus on the paper's rigorous theoretical foundation, conceptual novelty, and strong empirical results, though reviewer C2Qr argued the semantic affinity matrix does not capture genuine structural flow and critiqued the pipeline's computational overhead, while reviewers hCXB and jpHD initially raised concerns regarding anchor sensitivity and missing non-graph semantic baselines. During the author-reviewer discussion period, the authors provided extensive new experiments, including a non-FGW semantic baseline, causal structure ablations to directly address C2Qr's structural critique, and a highly effective graph-medoid anchor selection algorithm that satisfied r13d's robustness concerns. While hCXB explicitly acknowledged all concerns were fully resolved but chose not to update their negative score, and C2Qr remained unconvinced regarding scalability despite the authors accurately noting that fact-extraction overheads are standard across contemporary long-form UQ baselines, the overall technical merits of the paper are demonstrably solid. I recommend acceptance because the methodological shift from isolated fact-checking to graph-based structural alignment is a valuable and sound contribution, and the authors' rigorous rebuttals convincingly demonstrated the framework's robustness; authors must ensure the graph-medoid anchor selection and limitations regarding factual ambiguity in nuanced domains are prominently integrated into the camera-ready manuscript.